# From Memorization to Parameter Interference:
# How Overtraining Experts Harms Model Merging

Stefan Horoi [1 2]   Guy Wolf [1 2]   Eugene Belilovsky [3 2]   Gintare Karolina Dziugaite [4 2]

## Abstract

Modern deep learning is increasingly characterized by the use of open-weight foundation models that can be fine-tuned on specialized datasets. This has led to a proliferation of expert models and adapters, often shared via platforms like HuggingFace and AdapterHub. Model merging has recently emerged as an effective way to leverage these existing resources, enabling the composition of capabilities from different model checkpoints. A natural pipeline has thus formed to harness the benefits of transfer learning and amortize sunk training costs: models are pre-trained on general data, fine-tuned on specific tasks, and then multiple checkpoints are merged to obtain a more capable model. A prevailing assumption is that improvements at one stage of this pipeline propagate downstream, leading to gains at subsequent steps. In this work, we challenge that assumption by examining how expert fine-tuning affects model merging. We show that long fine-tuning of experts that optimizes for their individual performance leads to degraded merging performance across vision and language modalities, multiple model scales, and both fully fine-tuned and LoRA-adapted models. We trace this degradation to the memorization of a small set of difficult examples that dominate late fine-tuning steps. This causes negative parameter interference and encodes knowledge that is forgotten during merging. Finally, we demonstrate that task-dependent aggressive early stopping strategies can significantly improve model merging performance. [1]

## 1. Introduction

The rise of open-weight foundation models, such as CLIP (Radford et al., 2021; Ilharco et al., 2021), T5 (Raffel et al., 2020) and the more recent Gemma (Gemma Team, 2025), Llama (Grattafiori et al., 2024) and DeepSeek (DeepSeek-AI, 2024), has caused a paradigm shift in the field of machine learning. Instead of training a model from scratch as was previously the norm, it is now increasingly common for practitioners and researchers alike to start with a pre-trained foundation model and then fine-tune it on a task of interest (Stanford-CRFM, 2021). This approach leverages the benefits of transfer-learning, leading to performance and robustness gains. The proposal of multiple parameter-efficient fine-tuning (PEFT) methods (Hu et al., 2022; Liu et al., 2022), which reduce the computational costs of fine-tuning and limit catastrophic forgetting by only updating a subset of the model parameters, further enables this approach. This has lead to a proliferation of different versions of these foundation models and of PEFT adapters, fine-tuned on a variety of downstream tasks, which are openly accessible on public model repositories such as Hugging Face (Wolf et al., 2020) and Adapter Hub (Pfeiffer et al., 2020).

Model merging has recently emerged as an effective way to leverage existing model checkpoints and adapters. Merging methods allow the combination of multiple fine-tuned versions of the same foundational model into one, preserving the size and therefore the computational and memory requirements of the original pre-trained model while infusing it with multiple new capabilities (Matena & Raffel, 2022; Jin et al., 2023; Ilharco et al., 2023; Yadav et al., 2023; Yu et al., 2024; Davari & Belilovsky, 2024). The advent of model merging techniques and open-source libraries for merging (Kandpal et al., 2023; Goddard et al., 2024) has had an important impact on the deep learning community, providing a simple, training-free way to create better models from already existing checkpoints and adapters. In the past year, many of the top performing models on HuggingFace's Open LLM Leaderboard (Beeching et al., 2023) have resulted from the merging of fine-tuned checkpoints (Yu et al., 2024).

A natural pipeline has therefore emerged to leverage the benefits of transfer-learning and amortize past sunk train-

---

[1]Université de Montréal [2]Mila – Québec AI Institute [3]Concordia University [4]Google DeepMind. Correspondence to: Stefan Horoi <stefan.horoi@mila.quebec>, Gintare Karolina Dziugaite <gkdz@google.com>.

*Proceedings of the 43rd International Conference on Machine Learning*, Seoul, South Korea. PMLR 306, 2026. Copyright 2026 by the author(s).

[1]Our code is publicly available at: https://github.com/shoroi/overtrained_merging.

ing costs: large models are *pre-trained* in an unsupervised fashion on large amounts of general, unlabeled data; these foundational models are then *fine-tuned*, potentially using PEFT techniques, on specialized datasets or tasks; finally these fine-tuned expert checkpoints or adapters are *merged* and combined to create more capable, often multi-task models.

A common assumption is that *increased performance at one stage of this pipeline will propagate downstream.* In other words, a stronger pre-trained model should yield a stronger fine-tuned model, and similarly, stronger fine-tuned experts should produce a stronger merged model. We challenge this assumption in this work by studying the following questions: *How does expert training affect model merging?* and *Do all capabilities and knowledge transfer equally well?*

We find that long fine-tuning that optimizes for expert performance can substantially hurt model merging, a phenomenon to which we refer as "overtraining" in the context of this paper. While overtrained experts might be better on their respective fine-tuning tasks, they lead to worse performance when merged. We validate this phenomenon across diverse settings, including merging fully fine-tuned and LORA-adapted models, in both vision and language domains and across different model families and sizes. We use tools from the data difficulty literature to link prolonged training to the memorization of hard examples. This memorization causes negative parameter interference, leading to hard examples being overwhelmingly forgotten during merging, while easy examples remain correctly classified While some recent work has hinted that undertraining experts can benefit merging performance (Pari et al., 2024; Zhou et al., 2025), our work provides a systematic analysis of this phenomenon, and demonstrates how simple early stopping strategies can significantly improve the efficacy of existing merging techniques. Our research introduces a critical new dimension to model merging, showing how careful expert training, and targeted checkpoint release can unlock improved performance.

Concretely, our contributions are the following:

- We show that overtraining full fine-tuned (FFT) models produces sub-optimal merges (Section 3.1), and that the negative impact is even stronger when using LoRA adapters for parameter-efficient fine-tuning (Section 3.2);

- We explain this phenomenon through the lens of data difficulty in Section 4, showing that later training steps are dominated by the memorization of a small fraction of difficult examples, which are predominantly forgotten during merging due to negative parameter interference.

- We show that task-dependent expert training duration can further improve model merging performance. We

propose early stopping as a general principle to encourage expert undertraining. Our early stopping strategies effectively adapt training duration per task and can recover optimal merging accuracy (Section 5).

**Conflict of Interest Disclosure**   The author G.K.D. is employed by Google, which led the development of the ViT, T5 and BERT models evaluated in this paper.

## 2. Preliminaries and methodology

### 2.1. Model merging

Model merging has recently gained a lot of popularity as a means to combine the abilities of multiple fine-tuned versions of the same pre-trained model into one, preserving the model architecture and size (Yang et al., 2026). Formally, a model merging method, $Merge$, takes the parameters $\theta_0$ of the pre-trained foundation model, and parameters $\{\theta_t\}_{t \in \mathcal{T}}$ of the multiple *experts*, which are fine-tuned models on each task $t$ from a set $\mathcal{T}$, and outputs the parameters of the merged model $\bar{\theta} = Merge(\theta_0, \{\theta_t\}_{t \in \mathcal{T}})$. A simple example of this combination step is averaging the different fine-tuned models' parameters:

$$\bar{\theta} = \tfrac{1}{|\mathcal{T}|} \textstyle\sum_{t \in \mathcal{T}} \theta_t. \tag{1}$$

Merging methods are generally motivated by the observation that fine-tuned models exhibit *linear mode connectivity*: their loss minima are connected by low-loss linear paths in parameter space (Frankle et al., 2020; Sharma et al., 2024). This property typically emerges because fine-tuned models share substantial portions of their training trajectories (Frankle et al., 2020; Neyshabur et al., 2020), and is therefore a key reason merging is expected to be feasible. Nonetheless, a common challenge in model merging is the observed performance degradation of the merged model $\bar{\theta}$ on individual tasks $t \in \mathcal{T}$, relative to the original fine-tuned model $\theta_t$. This phenomenon has been coined "interference", and a plethora of merging methods have been proposed to reduce interference when merging models and to preserve as much of the accuracy of the expert models as possible (Matena & Raffel, 2022; Jin et al., 2023; Yadav et al., 2023; Yu et al., 2024; Deep et al., 2024; Davari & Belilovsky, 2024). These methods have mainly focused on modifying the experts parameters $\{\theta_t\}_{t \in \mathcal{T}}$ or the respective *task vectors* $\{\tau_t\}_{t \in \mathcal{T}}$, where $\tau_t = \theta_t - \theta_0$, and / or changing the combination step. We consider 4 popular merging methods:

- **Average** simply averages the parameters of all fine-tuned models following Equation (1);

- **Task Arithmetic (TA)** (Ilharco et al., 2023) scales the sum of the task vectors by a tuned scalar $\lambda$, and adds it to the pre-trained model parameters, returning $\theta_0 + \lambda \sum_{t \in \mathcal{T}} \tau_t$;

- **TIES** (Yadav et al., 2023) prunes low magnitude parameters from each task vector, then only averages the remaining parameters if they have the same sign as the weighted majority;

- **DARE** (Yu et al., 2024) randomly prunes a fraction of each task vector parameters; the remaining parameters are rescaled based on the pruning fraction, and are combined as in TA.

- **Iso-C & Iso-CTS** (Marczak et al., 2025) operate on per-layer task matrices; Iso-C sums them and replaces the SVD singular value spectrum with its mean, while Iso-CTS additionally appends task-specific singular directions from the orthogonal complement of the common subspace.

- **TSV-M** (Gargiulo et al., 2025) computes a truncated SVD of each task matrix and whitens the singular vectors across tasks to decorrelate them before merging.

We review other popular methods in Appendix A and detail our hyperparameter tuning procedure for merging in Appendix B. Prior works have primarily focused on deriving new techniques to reduce interference while assuming fixed, standard fine-tuning protocols. The role of the fine-tuning procedure itself, particularly its duration, has received little attention, with some exceptions discussed in Section 6. Our work explicitly studies how expert training time affects mergeability.

## 2.2. Low-rank adaptation

Modern foundation models have tens, if not hundreds, of billions of parameters, making full fine-tuning impractical on typical hardware (Grattafiori et al., 2024; DeepSeek-AI, 2024; Gemma Team, 2025). Parameter-Efficient Fine-Tuning (PEFT) updates only a small subset of the parameters to ease the computational burden and curb catastrophic forgetting (Hu et al., 2022; Liu et al., 2022). Low-Rank Adaptation (LoRA) (Hu et al., 2022), has emerged as one of the most popular PEFT methods due to its simplicity and effectiveness. LoRA inserts two low-rank matrices $\mathbf{A}$ and $\mathbf{B}$ into selected linear layers of a model. If the input and output dimension at that layer are $n_{in}$ and $n_{out}$, LoRA uses a rank $r \ll \min(n_{in}, n_{out})$ to define matrices $\mathbf{A} \in \mathbb{R}^{r \times n_{in}}$ and $\mathbf{B} \in \mathbb{R}^{n_{out} \times r}$. The output of that layer then becomes $(\mathbf{Wx} + \mathbf{b}) + \frac{\alpha}{r}\mathbf{BAx}$ where $\alpha$ is a scaling hyperparameter. During fine-tuning, the original model is frozen and only the LoRA's $\mathbf{A}, \mathbf{B}$ matrices are updated.

**Merging LoRA adapters** At each layer, the weight update induced by LoRA is exactly $\Delta W = W_{\text{fine-tuned}} - W_{\text{pre-trained}} = \frac{\alpha}{r}\mathbf{BA}$. Consequently, standard merging techniques can be directly applied to LoRA-adapted models if the updates $\frac{\alpha}{r}\mathbf{BA}$ are added to the pre-trained weights or if they are directly used to compute the task vectors. Merging the LoRA $\mathbf{A}$ and $\mathbf{B}$ matrices separately is not recommended since this can lead to mismatched representation spaces resulting in poor performance (Stoica et al., 2025). Nevertheless, recent work has observed that merging LoRA-adapted models is harder than merging FFT models (Tang et al., 2024; Stoica et al., 2025), often leading to significant performance degradation.

## 2.3. Data difficulty

In this work, we use data difficulty scores to identify which knowledge is transferred during merging and to relate merging performance to training dynamics and memorization. Specifically, we use the EL2N score introduced by Paul et al. (2021) which measures the norm of the error vector, i.e. the predicted class probabilities minus the one-hot label encoding. For a training example $x$ with one-hot label $y$, the EL2N score is defined as $\mathbb{E}\|p(\theta, x) - y\|_2$, where $p(\theta, x)$ denotes the model's predicted class probabilities for $x$ under parameters $\theta$.

Prior work has examined how individual data points influence neural network training dynamics and properties such as generalization, memorization, and privacy, leading to the development of various data difficulty scores (Kwok et al., 2024). These scores aim to quantify an intrinsic characteristic of the data, namely *data difficulty*, which captures how easy or hard individual examples are to learn. Easy examples typically exhibit common, easily learnable features, whereas hard examples often possess idiosyncratic structure or even noisy labels. Such scores have been successfully used for data pruning, with past work showing that large fractions of easy examples can be removed with little effect on performance, while pruning a small fraction of the hardest examples can improve generalization by eliminating outliers with uncommon features (Toneva et al., 2019) or mislabeled data (Paul et al., 2021). Moreover, Sorscher et al. (2022) showed that appropriate data pruning can yield better-than-power-law error scaling with dataset size.

A natural relationship exists between data difficulty and deep learning generalization and memorization. For instance, Sorscher et al. (2022) found a 0.78 Spearman rank correlation between EL2N scores (Paul et al., 2021) and the memorization score presented by Feldman & Zhang (2020). These observations suggest that correctly classifying difficult examples often requires memorization, and that a certain degree of memorization is therefore necessary for achieving near-optimal generalization. This relationship between memorization and generalization has been further substantiated with theoretical results in simpler settings (Attias et al., 2024; Feldman, 2020).

## 2.4. Models and datasets

**Vision domain** We evaluate merging performance in a standard vision benchmark setting using the official code-

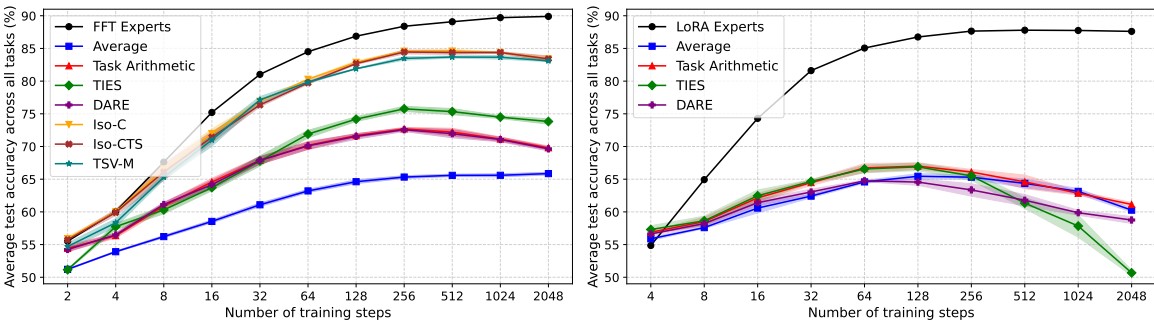

*Figure 1.* Average test accuracy across all 8 vision classification tasks for fully fine-tuned (**right**) and LoRA-adapted (**left**) ViT-B-32 models. We plot the average accuracy of the expert models evaluated on their respective tasks as well as merging accuracies for multiple methods. Shaded regions show mean±std over 3 random seeds.

base from (Ilharco et al., 2023): a CLIP (Radford et al., 2021) pre-trained ViT-B-32 model (Dosovitskiy et al., 2021) is fine-tuned on 8 image classification tasks: Cars (Krause et al., 2013), DTD (Cimpoi et al., 2014), EuroSAT (Helber et al., 2019), GTSRB (Stallkamp et al., 2012), MNIST (Deng, 2012), RESISC45 (Cheng et al., 2017), SUN397 (Xiao et al., 2010) and SVHN (Netzer et al., 2011). The fine-tuning is done with a batch size of 128, the AdamW optimizer (Loshchilov & Hutter, 2019; Paszke et al., 2019) and a learning rate of 1e-5. We use a learning-rate scheduler with linear warm-up for the first 10% of steps, followed by cosine annealing. When evaluating merged models, we use the corresponding frozen classification head for each task.

**Language domain** For our natural language processing (NLP) experiments, we adopt the setting of the TIES paper (Yadav et al., 2023) and use their released code. We use pre-trained T5-Base models (Raffel et al., 2020) which we fine-tune on 7 tasks: QASC (Khot et al., 2020), WikiQA (Yang et al., 2015) and QuaRTz (Tafjord et al., 2019) for question answering; PAWS (Zhang et al., 2019) for paraphrase identification; Story Cloze (Sharma et al., 2018) for sentence completion and Winogrande (Sakaguchi et al., 2020) and WSC (Levesque et al., 2012) for coreference resolution. We use the AdamW (Loshchilov & Hutter, 2019) optimizer with a batch size of 256, a constant lr of 0.0001 and no weight decay. bfloat16 mixed precision training is used to reduce GPU utilization.

**Evaluation** For all our experiments we report the raw, un-normalized test accuracy averaged across the multiple considered tasks. We chose not to use the popular *normalized accuracy* metric because the set of experts being merged here differs across experiments, which also changes the normalization factor and makes comparisons inconsistent. A more detailed justification is provided in Appendix C. Our experiments are ran using the PyTorch (Paszke et al., 2019) and HuggingFace (Wolf et al., 2020) open source machine learning frameworks on an Nvidia Quadro RTX 8000 GPU with 48GB of memory.

## 3. Longer fine-tuning hurts model merging

In this section, we present results challenging the common assumption that better fine-tuned models lead to better merging results. We show that overtrained experts lead to worse merged models for both FFT and LoRA adaptation.

### 3.1. Merging fully fine-tuned models

While a multitude of model merging methods have been proposed, the influence of the fine-tuning procedure itself on merging remains understudied. Most prior works have used similar fine-tuning protocols, typically training for a fixed 2000 steps in the vision setting described in Section 2.4. Instead of proposing yet another model merging method, we take a look at how the number of training iterations affects merging. We fine-tune our vision and NLP models for varying number of training steps $s \in \{2, 4, 8, 16, 32, 64, 128, 256, 512, 1024, 2048\}$ on every considered dataset. Each merge combines either 8 vision or 7 NLP experts (one per task) all trained for the same duration.

Figure 1 (left) shows that, except for Average, all methods achieve better merging performance when the ViT experts are trained for less than the commonly used 2000 steps. TA, TIES, and DARE yield models with ∼3% higher accuracy at 256 steps compared to 2048, a gain comparable to the 3.4% gap between TA and TIES at 2048 steps. The SVD-based methods are more robust to expert overtraining. They nonetheless achieve optimal performance earlier in training; Iso-CTS peaks at 256 steps with an accuracy of 84.5%, while both Iso-C and TSV-M peak at 512 with accuracies of 84.6% and 83.7% respectively. Their accuracies drop by $0.6 - 1.2\%$ at 2048 steps, with TSV-M being the most robust.

The same qualitative trend holds in the NLP setting (Figure 2 left), with all merging methods except for Average attaining peak merging performance at 256 steps. Further training leads to a drop in merging performance of over

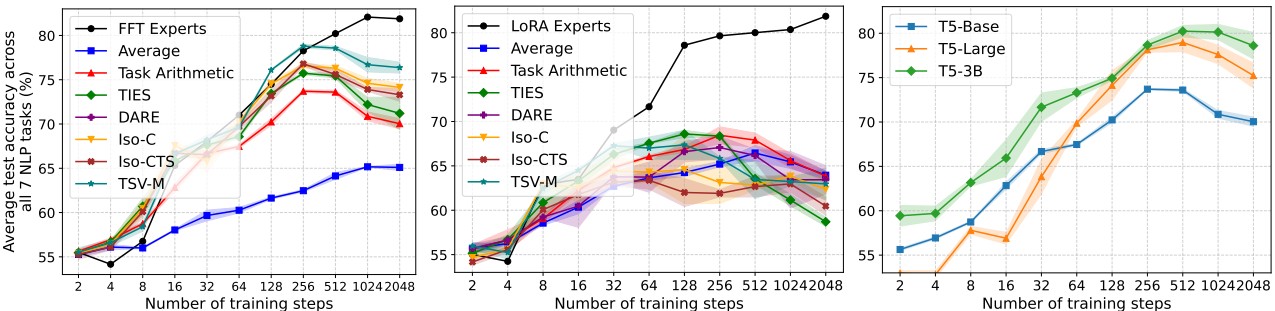

*Figure 2.* Average test accuracy across all 7 NLP tasks for fully fine-tuned (**left**) and LoRA-adapted (**center**) T5-Base models. We plot the average accuracy of the expert models evaluated on their respective tasks as well as merging accuracies for multiple methods. **Right:** Task Arithmetic merging accuracy for different T5 model sizes. Shaded regions show mean±std over 3 random seeds.

2.4% for all merging methods. The sparsity-based methods, TIES and DARE, are the most affected by expert overtraining, with drops in accuracy of 4.5% and 4.6% respectively at 2048 steps relative to 256. The SVD-based Iso-C and TSV-M are slightly more robust, both suffering from a 2.4% drop in performance from 256 to 2048 steps. TA and Iso-CTS suffer drops of 3.7% and 3.5% respectively. Notably, merging undertrained experts with TA outperforms merging experts trained for longer with TIES, showing that the cost of overtraining can outweigh the benefits of a better merging method. Average is the only method that does not degrade with longer training, reaching a peak accuracy of 65.2% at 1024 steps which is maintained at 2048 steps. However, Average consistently underperforms overall. Moreover, all the other merging methods show similar trends across training durations, suggesting that training length itself, rather than the merging method, plays a key role in merging performance.

**Better experts do not necessarily lead to better merging**
The black lines in the left and central panels of Figures 1 and 2 show the average accuracy of the expert models on their respective fine-tuning tasks. In both the vision and NLP settings, we observe that higher expert accuracy does not necessarily translate into better merging performance. In the vision setting, expert models trained for 256 steps achieve an average accuracy of 88.4%, which is 1.6% lower than at 2048 steps (90.0%). Nevertheless, merging after 256 steps yields models with approximately 3% higher accuracy than merging after 2048 steps for TA, TIES and DARE, and ∼ 1% higher for the SVD-based methods. The discrepancy is even more pronounced in the NLP setting. Expert accuracy improves from 78.2% at 256 steps to 81.9% at 2048 steps, a 3.7% gain, yet the merging accuracies of all methods drop by at least 2.4% over the same interval. We provide a per-task breakdown of these results in Appendix D and further experiments with ViT-L-14 and BERT models in Appendix E. The same "overtraining degrades merging" phenomenon is generally observed.

**Effect of model scale** In the right panel of Figure 2, we compare Task Arithmetic merging accuracy across different model sizes in the T5 family: T5-Base (220M parameters), T5-Large (770M), and T5-3B (3B). We observe that the same trend persists across scales: model merging performance peaks at an intermediate number of training steps before degrading with longer fine-tuning. Additional merging results are provided in Appendix F.

### 3.2. Merging LoRA adapters

We now extend our previous results to the highly relevant setting of merging LoRA adapters. We find that long training of LoRA experts hurts merging performance even more than in the FFT case. We add LoRA adapters at every linear layer of the original ViT-B-32 and T5-Base models. We use LoRA rank $r = 8$, scaling parameter $\alpha = 32$ and learning rates 1e-4 and 5e-4 for the ViT and T5 models respectively. We train the LoRAs for different number of steps $s$ to evaluate the impact of training duration on accuracy and mergeability. The parameters of the base model are kept frozen.

**Overtraining severely impairs LoRA merging** The center panels of Figures 1 and 2 show expert and merging accuracies for our vision and NLP LoRA models, respectively. For the ViT models, merging performance peaks at 128 training steps (64 for DARE), with accuracies ranging from 64.7–67.0% across all four methods. Although further training improves expert accuracy by about 1%, it significantly degrades merging performance, with accuracy drops of 5–6% for Average, TA, and DARE, and over 16% for TIES. We omit the SVD-based merging results for LoRA-adapted ViT models, since they all performed worse than the pretrained model in our experiments. In the NLP setting, methods reach peak merging performance at different but consistently short durations, between 64 and 512 steps: Iso-CTS peaks earliest at 64 steps (63.4%), TIES, Iso-C and TSV-M at 128 (68.6%, 64.6% and 67.4%), TA and DARE at 256 (68.5% and 67.1%), and Average latest at 512 (66.5%). Expert models, however, continue to improve, reaching an

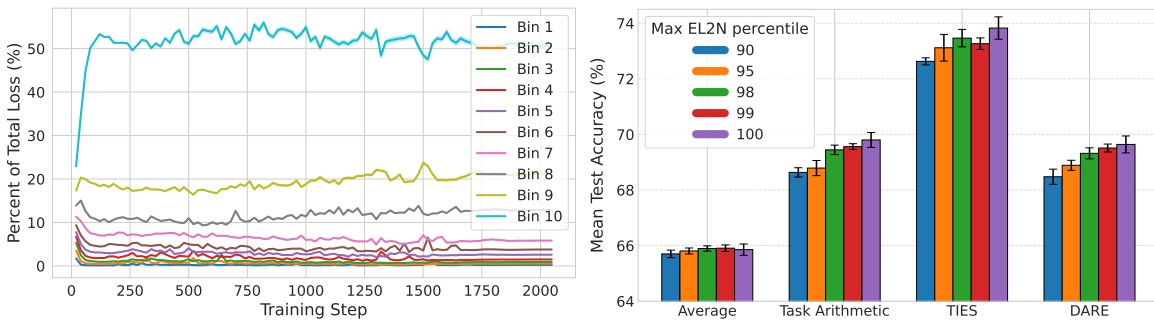

*Figure 3.* **Left:** Percentage of total loss for examples in different data difficulty bins. Bin 1 represents 10% easiest examples (lowest EL2N scores), bin 10 represents 10% hardest examples (highest EL2N scores). Mean across all 8 vision datasets shown. **Right:** Merging accuracy for experts trained without the hardest examples. Experts are trained on data with EL2N scores from percentile 0 to varying max percentiles in $\{90, 95, 98, 99, 100\}$.

average accuracy of 81.9% at 2048 steps. Despite this, merging at 2048 steps harms every method, with drops of 2.2–2.9% for Iso-C, Average and Iso-CTS, 3.6–4.6% for DARE, TSV-M and TA, and a striking 9.9% for TIES. In Appendix G, we examine the impact of LoRA rank and show that higher ranks lead to smaller performance degradations.

**Extension to Model MoErging** Our findings on expert overtraining also extend to model MoErging, an alternative "upcycling" strategy that combines adapters into modular mixture-of-experts (MoE) layers rather than merging parameters directly (Yadav et al., 2025; Ostapenko et al., 2024). In Appendix K, we show that MoE-fied models initialized with overtrained LoRA experts reach approximately 2% lower final multi-task accuracy than models initialized with undertrained experts, even after continued multi-task training of the router and expert modules. This indicates that the negative impact of expert overtraining is not specific to parameter merging, but reflects a more general sensitivity of model upcycling methods to expert training dynamics.

## 4. Why does overtraining harm merging?

In this section, we use tools from the data difficulty literature to explain why overtraining is detrimental to model merging. This allows us to make a series of empirical observations linking prolonged training to the memorization of hard examples and to increased parameter interference.

For all the training examples from the 8 considered image classification tasks we compute the EL2N data difficulty scores early in fine-tuning, after only 32 steps, across 10 different seeds (different models than the ones we merge). We note that despite being computed early in training, the EL2N scores aim to estimate an intrinsic characteristic of the data, independent of model training. To facilitate analysis, we group the training examples into 10 bins according to their EL2N scores, the 10% of examples with the lowest EL2N scores (the easiest examples) are in bin 1 and so on.

**Observation 1: Later training stages are driven by the memorization of hard examples.** In the left panel of Figure 3 we show the relative loss of the training examples during training. We observe that easy examples, which have more common features, are learned very early in training. **The remaining of training is driven largely by the loss of the difficult examples**, with the top 10% of hardest examples accounting for over 50% of the total loss after the first 100 steps. In Appendix H we quantify memorization of examples in each data difficulty bin using three metrics: *change in margin* (gap between true class probability and maximum probability among other classes), *change in loss*, and *predictive distribution shift* (L1 norm of the difference between predicted probabilities). Comparing ViT-B-32 models trained for 256 and 2048 steps, all metrics show changes orders of magnitude larger for the hardest examples (bin 10) than for the easiest (bin 1), despite only modest accuracy gains on the test set (1.6% on average) and on the training set (4%). This suggests that **memorization of hard examples occurs during the later stages of training**.

To isolate the causal influence of individual examples, we adopt the memorization framework of Feldman (2020), which compares models trained with and without a given example. For each vision dataset, we trained 40 models with different random seeds. In each run, we held out 90 training examples: 30 from the 100 easiest, 30 from the 100 hardest, and 30 from the 100 closest to the median difficulty (as ranked by EL2N scores). Under this design, each example in the three bins (easiest, middle, hardest) has an inclusion rate of $\sim$70% across runs. Because inclusion is randomized across seeds, this comparison estimates the effect of including $x$ in training, free of confounds. This lets us compute the (Feldman) memorization score (Feldman, 2020) for each example throughout training:

$$\mathrm{mem}(x, t) = \mathbb{E}[\, c_t(x) \mid x \in S \,] - \mathbb{E}[\, c_t(x) \mid x \notin S \,],$$

where $c_t(x)$ is the confidence the model assigns to the correct class of $x$ at step $t$, and the expectations run over the

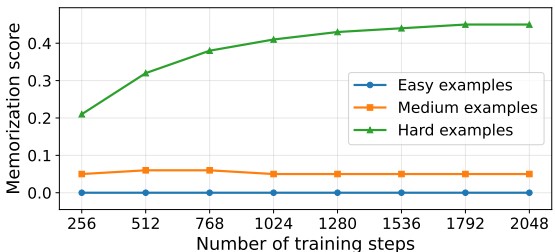

*Figure 4.* Memorization scores ([Feldman](#), 2020) (mean, aggregated across vision datasets) as a function of training step, stratified by example difficulty (easy, medium, hard).

seeds whose training set $S$ included or excluded $x$. The score measures how much the model's confidence on $x$ owes specifically to having trained on $x$: a large value indicates an example-specific influence that the model can acquire only by fitting $x$ itself. A high score is not in itself a sign of harmful overfitting: under the long-tail account of [Feldman](#) ([2020](#)), fitting rare or atypical examples can be necessary for good generalization.

[Figure 4](#) reports these scores aggregated across datasets. Easy examples are essentially never memorized (scores $\leq 0.01$ at every step), and middle-difficulty examples show only small, stable memorization ($\approx 0.05$) that does not grow with training. Hard examples behave differently: their memorization accumulates throughout training, more than doubling from 0.21 at step 256 to 0.45 at step 2048. Memorization is therefore concentrated on the hardest examples and builds up over the late training stages we find harmful to merging. Holding out a hard example sharply reduces the model's confidence on it, while holding out an easy or medium example has almost no effect on the model's behavior.

**Observation 2: Later training stages result in idiosyncratic parameter updates causing increases in parameter interference.** As discussed in [Section 2.3](#), hard examples represent outliers with uncommon features or noisy labels. Therefore, it stands to reason that the memorization of such examples would also yield idiosyncratic parameter updates that don't generalize across tasks. In [Appendix I](#) we estimate the amount of parameter interference between the task vectors obtained with various training durations using four different metrics: *sign conflict percentage*, *parameter overlap percentage*, *magnitude ratio* and *per-parameter variance*. All of these parameter interference scores increase with training duration, confirming that **longer training yields idiosyncratic parameter updates leading to more parameter interference.** Since parameter interference is a well accepted explanation to the performance degradation observed when merging ([Yadav et al.](#), [2023](#)), **this directly explains our main observation that longer training negatively impacts model merging**.

To directly link these parameter updates to the memorization of hard examples, we take a look at which examples are *forgotten* during merging, i.e. training examples correctly classified by the expert models but incorrectly classified after merging. [Figure 5](#) shows that hard examples are disproportionately forgotten during merging, with over 50% of forgotten data points in the top 30% in terms of data difficulty. This confirms that parameter updates from prolonged training are idiosyncratic and driven by memorization of hard examples, and that merging destroys some some of these learned features due to parameter interference.

**Observation 3: Difficult examples are still necessary for good merging generalization.** Inspired by work showing that removing difficult examples from training can aid generalization ([Toneva et al.](#), [2019](#); [Paul et al.](#), [2021](#)), we investigate how this affects merging performance. We remove the top 1, 2, 5, or 10% most difficult examples from training and report merging results in the right panel of [Figure 3](#). However, we find that the best merging results are achieved when all data is used and that removing difficult examples consistently hurts performance. This expands upon past work showing that some memorization is necessary for close-to-optimal generalization ([Feldman](#), [2020](#); [Feldman & Zhang](#), [2020](#); [Attias et al.](#), [2024](#)) by demonstrating that **some amount of memorization is also necessary for close-to-optimal merging performance.**

## 5. Aggressive early stopping improves merging results

Our core finding that overtraining experts harms merging performance naturally motivates the use of early stopping to mitigate this effect. We hypothesize that early stopping during fine-tuning will yield stronger merging results, as it both shortens training duration and automatically adapts the stopping time for each task. In this section we investigate this hypothesis and find that merging can indeed be improved by early stopping, even when expert-level performance is lower. Our goal here is not to establish the optimality of a single strategy, but to propose early stopping as a general principle for mitigating the negative effects of overtraining in model merging and to introduce practical, automated variants that follows established best practices.

### 5.1. Vision — Linear warm-up and adaptive decay

The learning rate scheduler used in [Section 3](#) for ViT models, a linear warm-up followed by cosine decay, is a standard choice in vision training and recent merging work, where warm-up provides early stability and decay supports smooth convergence. Our early stopping strategy for these models retains this warm-up/decay structure while making it adaptive to the shorter, task-dependent training lengths induced by early stopping. We achieve this by pairing a 50-step linear warm-up phase with a "reduce learning rate on plateau"

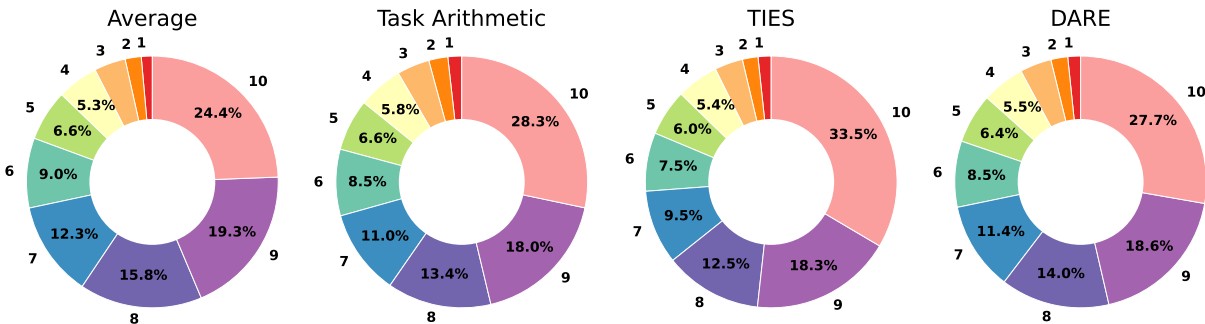

*Figure 5.* Proportion of forgotten examples in each data difficulty bin for three different model merging methods. Bin 1 represents 10% easiest examples (lowest EL2N scores), bin 10 represents 10% hardest examples (highest EL2N scores). Hard examples are overwhelmingly forgotten when merging with all methods, with the 30% hardest examples representing over 50% of forgotten examples.

phase that gradually decreases the learning rate when validation accuracy stagnates. We compute the validation accuracy every 5 training steps and multiply the learning rate by 0.5 when it has not improved for 3 consecutive rounds. Once the learning rate falls below a threshold of 1e-7, training is stopped. We fine-tune FFT and LoRA models on the 8 considered vision tasks using peak learning rates of 1e-5 and 1e-4 respectively. Pseudocode for this LR scheduler and early stopping strategy is provided in Appendix J.

In Table 1 we compare the merging of early stopped experts to two baselines from Section 3: merging "overtrained" models trained for 2048 steps and merging the checkpoints that achieved the highest accuracy among all training durations (same duration across tasks). We see that the models trained using our simple task-dependent early stopping strategy yield merges that are better than those of overtrained models and as good, if not better, than the best merged experts obtained with a common stopping time even though the experts are on worse on their respective tasks. Early stopping seems to work especially well for LoRA adaptation, yielding results on average better than the best ones from Section 3.2.

### 5.2. Language — Stopping when validation accuracy plateaus

Our T5 models from Section 3.1 were trained with a constant learning rate, making the design of the early stopping criterion straightforward: we evaluate accuracy on a held-out validation set every `#steps` batches and stop training whenever the validation accuracy fails to improve for `#wait` consecutive evaluations. The checkpoint with the highest validation accuracy is selected as the expert. We study two variants of this strategy: **v0**, with `#steps`=100 and `#wait`=5, and a more aggressive **v1**, with `#steps`=50 and `#wait`=3. Results are presented in Table 2.

Experts obtained with v0 match the performance of models trained for 1024 or 2048 steps, while v1 yields slightly lower accuracy. Importantly, both strategies achieve these results with far fewer training iterations, on average only

485 steps for v0 and 269 steps for v1. We also note high variance for the number of steps across tasks. Despite the reduced training time, the merging performance improves substantially. Task Arithmetic and TIES both benefit: across seeds, both early stopping strategies produce merges that are roughly 4% more accurate than merges from experts trained for 1024 or 2048 steps. Moreover, the merging results for early stopped models are even superior to the best results from Section 3.1 where the single best stopping point is selected commonly for all tasks. Average merging is the only method which doesn't seem to benefit from early stopping.

## 6. Related work

**Special fine-tuning procedures for better merging** Several recent works study how modifications to fine-tuning can improve merging performance. However, these approaches adjust the fine-tuning procedure itself, for example by updating only selected submodules (Jin et al., 2025) or parameters (Iurada et al., 2025), using sharpness-aware optimization (Lee et al., 2025), or employing linearization-based updates (Tang et al., 2024; Ortiz-Jimenez et al., 2023). In contrast, our work analyzes how *standard* fine-tuning protocols, without merging-specific adjustments, affect downstream merging performance, providing a complementary perspective on merging behavior under the most commonly used training procedures.

**Expert training time** Most model merging do not examine how expert fine-tuning affects downstream merging, with two notable exceptions. Zhou et al. (2025) show that the effectiveness of task-vector based approaches is largely driven by first-epoch gradients and propose alternating 1-epoch fine-tuning and merging. While they note that less training can improve accuracy, they only test 1, which can yield either overtrained or undertrained experts depending on dataset size. Closely related, Zhou et al. (2026) prove that, under idealized assumptions (notably full-batch gradient descent), a single-epoch task vector is ex-

*Table 1.* Experts and merging accuracy (%) for the overtrained, optimal and early stopped ViT-B experts. Mean and standard deviation across 3 random seeds shown. Bold indicates the best result per column within each fine-tuning style (FFT or LoRA); underline indicates second best.

|  | Experts | Average | Task Arithmetic | TIES | DARE |
|---|---|---|---|---|---|
| FFT 2048 steps | **89.9**±0.1 | **65.9**±0.2 | 69.8±0.2 | 73.8±0.4 | 69.6±0.3 |
| FFT best (# steps) | - | **65.9**±0.2 (2048) | **72.7**±0.2 (256) | **75.8**±0.4 (256) | **72.6**±0.3 (256) |
| FFT early stop | 87.9±0.2 | 64.5±0.1 | 72.6±0.5 | 74.7±0.3 | 72.5±0.5 |
| LoRA 2048 steps | 87.6±0.1 | 60.3±0.3 | 61.1±0.3 | 50.7±0.4 | 58.8±0.5 |
| LoRA best (# steps) | - | 65.4±0.4 (128) | 67.0±0.5 (128) | 66.9±0.6 (128) | 64.7±0.1 (64) |
| LoRA early stop | **87.9**±0.1 | **65.6**±0.4 | **68.0**±0.8 | **67.1**±0.6 | **65.3**±0.3 |

*Table 2.* Merging accuracy (%) and number of training steps for the overtrained, optimal and early stopped T5 experts. Mean and standard deviation across 3 random seeds shown. Bold indicates the best result per column; underline indicates second best.

|  | # steps | Experts | Average | Task Arithmetic | TIES |
|---|---|---|---|---|---|
| 1024 steps | 1024 | **82.4**±0.3 | **65.2**±0.1 | 70.9±0.6 | 72.2±1.0 |
| 2048 steps | 2048 | 82.0±0.5 | 65.0±0.4 | 70.1±0.7 | 72.2±1.4 |
| Best (# steps) | – | – | **65.2**±0.1 (1024) | 73.7±0.2 (256) | 75.7±0.2 (256) |
| Early stop v0 | 485±350 | 82.3±0.6 | 63.4±0.4 | **75.0**±0.3 | 76.9±0.5 |
| Early stop v1 | 269±199 | 81.3±0.4 | 62.8±0.4 | 74.0±0.1 | **77.0**±0.3 |

actly the negative task-loss gradient up to the learning rate, recasting task arithmetic as approximate multitask learning. This equivalence weakens over subsequent epochs. Our work is complementary, empirically quantifying how prolonged expert training degrades merging across model architectures, datasets, adaptation regimes, and merging methods, and yielding concrete, practitioner-facing guidance on training duration beyond what existing theory offers. Pari et al. (2024) observe representational incompatibilities when merging highly specialized experts, but study only two-model merges and propose bypassing merging altogether by using MoErging. To our knowledge, we are the first to systematically link expert training duration to downstream merging outcomes, analyze merging through example difficulty, and propose an early-stopping strategy that adapts to dataset heterogeneity. Finally, although TIES Merging (Yadav et al., 2023) uses early stopping, it is only used to avoid expert overfitting and its effect on merging is not mentioned.

**Overtraining in pre-training** Analogous to our work, others have studied how scaling pre-training impacts downstream fine-tuning. In a large-scale vision study, Abnar et al. (2022) found that as pre-training accuracy improves, fine-tuning saturates. Recently, Springer et al. (2025) show that over-training LLMs during pre-training can harm fine-tuned in- and out-of-distribution performance.

## 7. Conclusion

In this paper, we challenged the assumption that better fine-tuned experts yield better merging performance. Across multiple merging methods, model families and sizes, and for both fully fine-tuned and LoRA-adapted models, we found that optimal merging occurs well before full convergence, often when experts are less accurate on their original tasks. We attribute this to a shift in training dynamics: as fine-tuning progresses, training becomes dominated by memorization of difficult examples, leading to idiosyncratic parameter updates and negative parameter interference. Finally, we show that simple early stopping strategies mitigate overtraining and can even yield better merging performance.

Our findings have important implications for the sharing of model versions and adapters and the evaluation of merging pipelines. **Publish intermediate checkpoints:** Releasing not only final but also intermediate checkpoints is crucial, as the best merging point may precede convergence. In practice, even a single intermediate checkpoint, extracted when validation accuracy starts to plateau, is likely sufficient for achieving close-to-optimal merging performance, as supported by our early stopping experiments. **Prioritize early-stopped experts:** When training experts in-house, aggressive early stopping can outperform convergence for downstream merging. Since merging reuses checkpoints and amortizes sunk costs, our findings can help reduce the future computational and environmental footprint of training AI models.

## Acknowledgments

We are grateful to Ghada Sokar, Xiaofeng Zhang and Peter Plantinga for their valuable feedback on this paper. This work was partially funded by the Google-Mila grant; the Natural Sciences and Engineering Research Council of Canada (NSERC) CGS D 569345 - 2022 scholarship [S.H.]; FRQNT-NSERC grant 2023-NOVA-329125 [E.B. & G.W.]; Canada CIFAR AI Chair, NSF DMS grant 2327211 and NSERC Discovery grant 03267 [G.W.]. This work is also supported by resources from Compute Canada and Calcul Quebec. The content is solely the responsibility of the authors and does not necessarily represent the views of the funding agencies.

## Impact statement

This paper presents work whose goal is to advance the field of Machine Learning, specifically research on model merging and the reuse of fine-tuned expert models. A practical benefit of our findings is the potential reduction in computational and environmental costs: aggressive early stopping and the release of intermediate checkpoints can improve merging performance while requiring substantially less training. More broadly, improving the reliability of model merging can enable more efficient reuse of open-weight models and reduce duplication of training effort. The ethical and societal risks associated with this work are not substantially different from those of other advances in machine learning methodology, and we do not identify specific negative consequences that warrant special attention.

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

## A. Additional related work

The simplest approach, parameter averaging, was shown to lead to better generalization when used on checkpoints from the same training trajectory (Izmailov et al., 2018) and was popularized in federated learning with FedAvg (McMahan et al., 2017). Recently, parameter averaging was also shown to be useful in the context of robust fine-tuning (Wortsman et al., 2022) and to obtain better pre-trained models (Choshen et al., 2022). When merging multiple fine-tuned versions of the same pre-trained model, Fisher-weighted averaging (Matena & Raffel, 2022) and related methods improve upon this simple averaging by adjusting per-parameter contributions (Jin et al., 2023; Tam et al., 2024). Task arithmetic based methods rely on the computation of task vectors, which are then summed, scaled and added back to the pretrained model (Ilharco et al., 2023) to give it multi-task capabilities. Pruning the task vector parameters (Yadav et al., 2023; Davari & Belilovsky, 2024; Yu et al., 2024; Deep et al., 2024) and selectively combining them to reduce negative interference (Yadav et al., 2023) further benefits performance. Sharma et al. (2024) explores merging of experts that were trained from different or poorly performing pre-trained models.

## B. Tuning merging hyperparameters

Several merging methods require careful hyperparameter tuning to achieve optimal performance. In particular, Task Arithmetic, TIES, and DARE each apply a scaling factor $\alpha$ to their task-vector sums before adding them to the pretrained weights; TIES and DARE additionally specify a percentage $k$ of weights to retain after pruning. As is standard, we select the best $\alpha$, $k$ values by maximizing merging accuracy on a held-out validation set. All merging accuracies reported in the main text are evaluated on the test set using hyperparameters selected via validation performance. We followed the hyperparameter configurations from the original papers (Ilharco et al., 2023; Yadav et al., 2023; Yu et al., 2024), adjusting them as needed to optimize performance in our experimental settings.

**Crucially, we perform tuning of merging hyperparameters for each individual merging experiment, i.e. the tuning is done for each different number of training steps and each different random seed across all our settings.**

**Vision setting:** Following (Ilharco et al., 2023), we reserve 10% of the training data for validation and train the ViT models on the remaining 90%. We tune the following hyperparameter values using the validation set:

- **Task Arithmetic:** $\alpha \in \{0.05, 0.1, \ldots, 1\}$
- **TIES:** $\alpha \in \{0.5, 0.6, \ldots, 1.5\}$ and $k \in \{10, 20, 30\}$
- **DARE:** $\alpha \in \{0.05, 0.1, \ldots, 0.55\}$ and $k \in \{10, 20, 30\}$
- **Iso-C:** $\alpha \in \{1.0, 1.2, \ldots, 6.0\}$
- **Iso-CTS:** $\alpha \in \{1.0, 1.2, \ldots, 6.4\}$, the common subspace fraction is kept at 0.8 as per the authors' recommendation (Marczak et al., 2025).
- **TSV-M:** $\alpha \in \{0.4, 0.5, \ldots, 3.0\}$

**NLP setting:** We adopt the validation splits from (Yadav et al., 2023) and evaluate the following hyperparameter values:

- **Task Arithmetic:** $\alpha \in \{0.1, 0.2, \ldots, 1\}$
- **TIES:** $\alpha \in \{0.8, 0.9, \ldots, 2.1\}$ and $k \in \{10, 20, 30\}$
- **DARE:** $\alpha \in \{0.1, 0.2, \ldots, 2.1\}$ and $k \in \{10, 20, 30\}$
- **Iso-C:** $\alpha \in \{2.0, 2.2, \ldots, 6.0\}$
- **Iso-CTS:** $\alpha \in \{2.4, 2.6, \ldots, 6.4\}$, the common subspace fraction is kept at 0.8 as per the author's recommendation (Marczak et al., 2025).
- **TSV-M:** $\alpha \in \{0.1, 0.2, \ldots, 3.0\}$

## C. Using the raw, un-normalized accuracy

The *normalized accuracy* is a very common metric used to compare model merging methods (Ilharco et al., 2023; Yadav et al., 2023). However, because the normalized accuracy depends on both the merged model's performance and that of the experts, it isn't suitable for settings like ours where different sets of experts are used and compared.

The core issue is that normalized accuracy, defined as (merged_accuracy / expert_accuracy), is a relative metric

designed to compare different merging methods when the set of experts (the denominator) is fixed. Papers that propose a novel model merging method are justified in using this metric by the fact that they have a fixed set of experts and they are comparing merging methods, therefore only the numerator changes. In our study, the experts themselves are the primary variable, as their training duration and performance change in each experiment, therefore the denominator changes from one merging experiment to another. This creates paradoxical situations that make the metric misleading for our purposes. For example consider the following scenario:

- **Case 1 (Undertraining):** Experts trained for only a few steps have very low absolute accuracy (e.g., 60%). When merged, they interfere very little since they're all relatively close in parameter space to the zeroshot model, so the merged model also achieves around 60% accuracy. This yields a normalized accuracy near 100%, despite the models being bad at solving the considered tasks.

- **Case 2 (Optimal Training):** Experts trained for longer have high accuracy (e.g., 90%). Merging them results in a high-performing model with 85% absolute accuracy. However, the normalized accuracy is only $85/90 = 94.4\%$ due to negative interference caused by longer training.

Comparing the "useless" 100% from Case 1 with the "useful" 94.4% from Case 2 is meaningless. Absolute, unnormalized accuracy on the other hand allows for a fair and interpretable comparison of the final upcycled model's quality across different expert training durations.

## D. Per-task breakdown of expert and merging accuracies

Here we present per-task accuracy plots for the expert and merged models in both the vision (Figure 6) and NLP settings (Figure 7). While there is some variability to the magnitude of the overtraining effect across datasets and merging methods, the direction of the effect is fairly consistent: for most tasks and merging methods, experts trained for significantly fewer steps yield higher merging accuracy. Importantly, we do not rely on any single dataset or outlier task to support our conclusions: the phenomenon is robust across tasks, domains, and merging methods.

## E. Results with additional models

To further reinforce the generality of our results, we have also run experiments with other model architectures.

### E.1. Vision - ViT-L-14

In the vision setting we have fine-tuned CLIP (Radford et al., 2021) ViT-L-14 (Dosovitskiy et al., 2021) models on the same set of 8 image classification tasks introduced in Section 2. We used the same training hyper-parameters as for the ViT-B-32models. The results are shown in Figure 8. The same phenomenon can be oserved for the larger ViT-L-14 models, while the average expert accuracy keeps increasing throughout training, the average merging accuracy peaks early in training. Task Arithmetic reaches a peak accuracy of 85.6% at 512 steps before decreasing to 84.0% at 2048 steps. TIES reaches a peak accuracy of 87.7% at 512 steps of training before decreasing to 86.2% at 2048 steps. DARE also peaks at 512 steps with an average accuracy of 85.6% before decreasing to 84.0% at 2048 steps. Lastly, Average merging again seems more robust to the negative effects of overtraining on model merging, as in our main results. It continues improving with training reaching a peak accuracy of 79.4% at 2048 steps, only slightly better than the 79.1% achieved at 512 steps. However Average merging yields significantly worse results than all other considered methods, regardless of the number of fine-tuning steps.

### E.2. NLP - BERT Base

In the NLP setting we have fine-tuned BERT base (Devlin et al., 2019) language models on the same set of 7 natural language processing tasks introduced in Section 2. For training we used a constant learning rate of 2e-5, the remaining hyper-parameters being the same as for the T5 models. The results are shown in Figure 8. The same phenomenon can be oserved for the BERT models, while the average expert accuracy keeps increasing throughout training, the average merging accuracy peaks early in training. TIES reaches a peak accuracy of 69.0% at only 32 steps of training before decreasing to 58.5% at 2048 steps. Task Arithmetic reaches a peak accuracy of 68.6% at 128 steps before decreasing to 64.0% at 2048 steps. Lastly, even Average merging which seemed more robust to the negative effects of overtraining on model merging reaches a peak accuracy of 59.3% at 128 steps before decreasing to 58.6% at 2048 steps.

## F. Effect of model scale

In the right panel of Figure 2, we have investigated how model size influences the merging dynamics by comparing Task Arithmetic merging results across T5-Base (220M parameters), T5-Large (770M), and T5-3B (3B). In Figure 9 we also show the average expert accuracy on their respective tasks as well as the merging accuracies for Average, Task Arithmetic and TIES methods. The purpose of these experiments is to test whether the decrease in merging accuracy observed after extended fine-tuning in smaller models also occurs at larger scales. We find that the same phenomenon

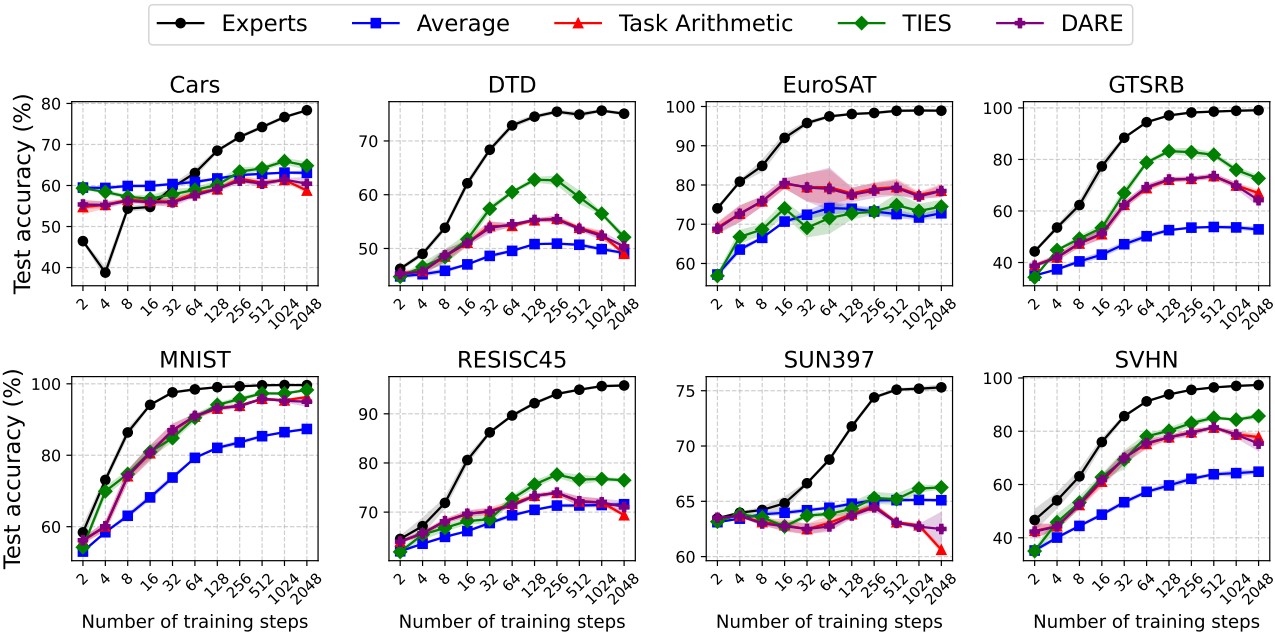

*Figure 6.* Per-task breakdown of expert and merging accuracies for our ViT-B-32 models trained on 8 image classification tasks. Mean and standard deviation across 3 random seeds shown.

persists: for both Task Arithmetic and TIES, merging accuracy peaks at an intermediate number of training steps and then degrades as fine-tuning continues, even though the absolute merging accuracy is generally higher for the larger models. Interestingly, Average merging appears robust to this degradation, but its overall accuracy remains comparatively low.

## G. Effect of LoRA rank

In this section, we examine how the choice of LoRA rank affects the degradation effect reported in the main paper. We find that increasing the LoRA rank mitigates the loss in merging accuracy that occurs as experts are trained for longer.

We fine-tune ViT-B-32 models on the eight image-classification tasks from Section 2.4, applying LoRA adapters to every linear layer while systematically varying the adapter rank $r$. We employ square-root scaling for the LoRA factor $\alpha$, choosing $(r, \alpha)$ in $\{(16, 45), (32, 64), (64, 90), (128, 128), (256, 181)\}$. The models are trained for varying number of steps $s \in \{8, 32, 128, 512, 2048\}$ to assess how training duration interacts with rank. When merging, we combine LoRA-adapted models with the same rank and trained for the same number of steps. The resulting accuracies are plotted in Figure 10.

Across all three merging methods (Average, Task Arithmetic, and TIES) increasing the LoRA adapter rank con-

sistently raises merging accuracy at every training duration. Moreover, higher ranks substantially attenuate the accuracy drop associated with extended training: as the number of fine-tuning steps grows, models with larger ranks exhibit smaller declines from their peak merging performance.

## H. Memorization Analysis of Early vs. Late Checkpoints

A central hypothesis in our work is that *longer finetuning drives models to increasingly memorize difficult training examples*. To test this, we compare CLIP ViT-B-32 models trained for **256** steps ("early") to models trained for **2048** steps ("late") across eight image classification datasets (Cars, DTD, EuroSAT, GTSRB, MNIST, RE-SISC45, SUN397, SVHN) with three random seeds each. Although both checkpoints show high training accuracy overall (100% on easy examples at both stages; 75.1% → 95.6% on hard examples from 256 to 2048 steps), memorization manifests not primarily in raw accuracy but in the *trajectory of prediction confidence, loss, and probability distributions*. We show that while accuracy on hard examples improves by 20.5 percentage points, the underlying memorization metrics exhibit changes that are *orders of magnitude* larger relative to easy examples, revealing the mechanism by which extended training drives memorization.

To quantify this, we compute three per-example metrics designed to detect late-stage memorization effects. All metrics are computed on the training set, and then aggregated

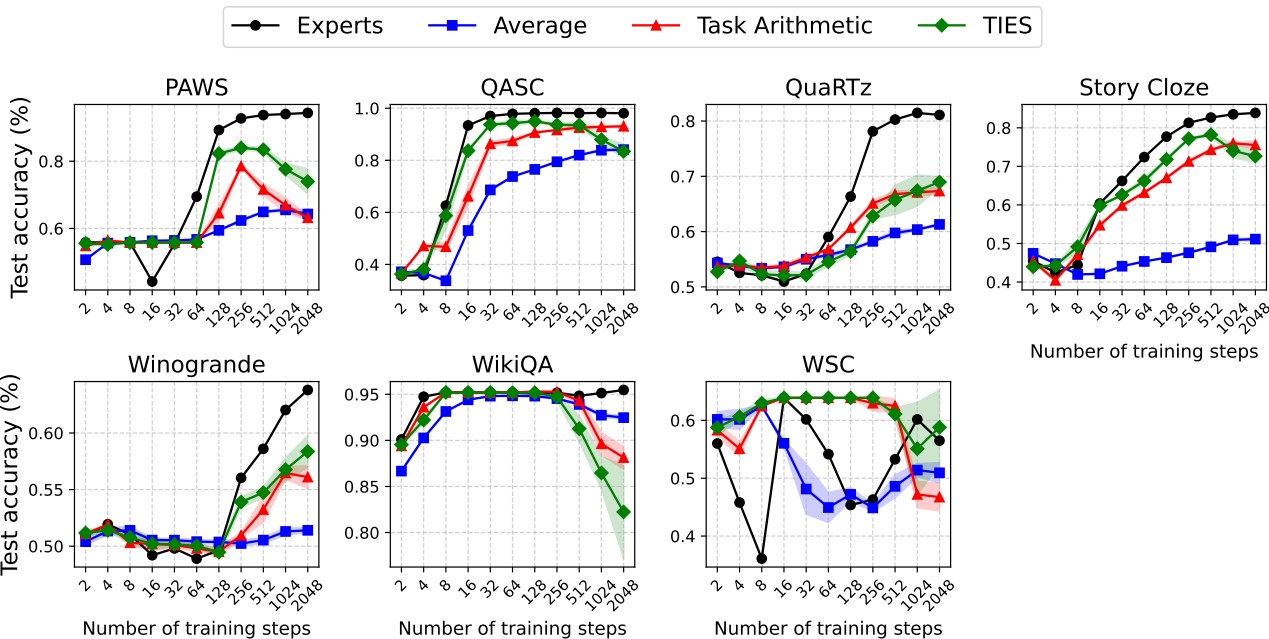

*Figure 7.* Per-task breakdown of expert and merging accuracies for our T5 models trained on NLP tasks. Mean and standard deviation across 3 random seeds shown.

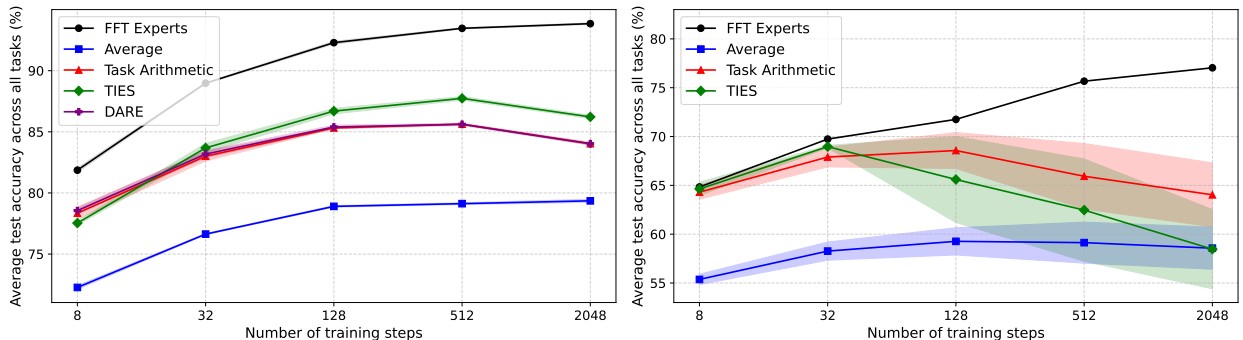

*Figure 8.* **(left)** Average test accuracy across all 8 vision tasks for fully fine-tuned ViT-L-14 models. **(right)** Average test accuracy across all 7 NLP tasks for fully fine-tuned BERT base models. We plot the average accuracy of the expert models evaluated on their respective tasks as well as merging accuracies for multiple methods. Shaded regions show mean±std over 3 random seeds.

into 10 difficulty bins per dataset using EL2N scores (Section 4). Each dataset is partitioned into 10 quantile-based bins, where bin 1 contains the easiest 10% of examples and bin 10 contains the hardest 10%. To prevent large datasets from dominating the averages, all results are first aggregated *per dataset* and then averaged across datasets (mean-of-means). For a training example with label $y$ at training step $t \in \{256, 2048\}$, we denote the predicted probability vector as $p^{(t)} = \mathrm{softmax}(z^{(t)})$ where $z^{(t)}$ are the logits.

### H.1. Memorization Metrics

**Change in margin:** $\Delta m = m^{(2048)} - m^{(256)}$ where $m^{(t)} = p_y^{(t)} - \max_{c \neq y} p_c^{(t)}$.
The margin measures the confidence gap between the true

class probability and the maximum probability assigned to any other class. A strong positive relationship between $\Delta m$ and difficulty indicates that the late model becomes disproportionately confident on hard examples.

**Change in loss:** $\Delta \ell = \ell^{(256)} - \ell^{(2048)}$ where $\ell^{(t)} = -\log p_y^{(t)}$.
This measures the drop in cross-entropy loss from early to late training. Large positive values indicate that the late model "forces down" the loss of examples that earlier checkpoints still struggled with, even when the top-1 prediction was already correct.

**Predictive distribution shift:** $\mathrm{shift} = \|p^{(2048)} - p^{(256)}\|_1$.
This $L_1$ distance quantifies how much the model's entire

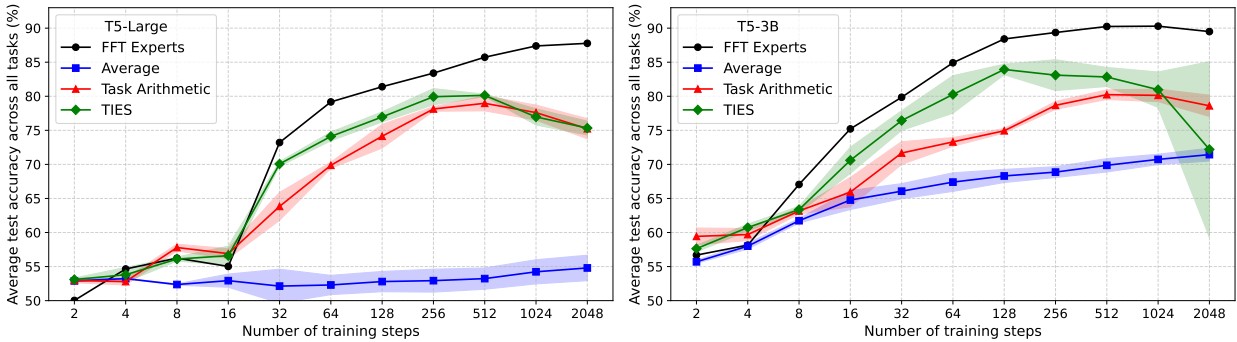

*Figure 9.* Average test accuracy across all 7 NLP tasks for fully fine-tuned T5-Large (**left**) and T5-3B (**right**) models. We plot the average accuracy of the expert models evaluated on their respective tasks as well as merging accuracies for multiple methods. Shaded regions show mean±std over 3 random seeds.

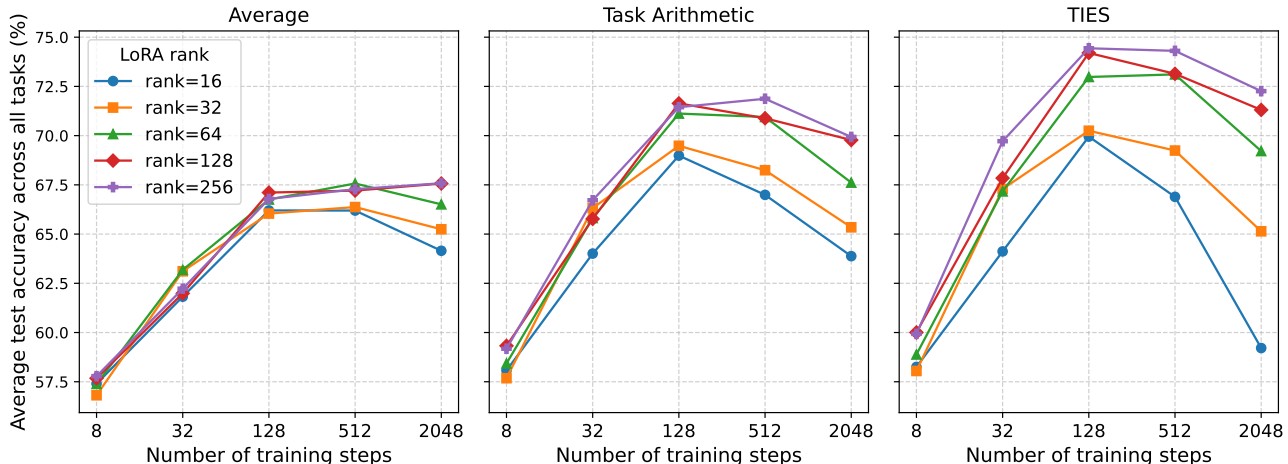

*Figure 10.* Average test accuracy across all 8 vision classification tasks as a function of the number of fine-tuning steps for different LoRA ranks and three merging methods. Each panel shows one method: Average (left), Task Arithmetic (center) and TIES (right). Colored solid lines and distinct markers denote the different LoRA adapter ranks. The x-axis is in $\log_2$ scale.

probability vector changes between checkpoints. Large shifts indicate substantial changes to the decision function, revealing targeted late-stage adjustments even when accuracy is unchanged.

### H.2. Results

Table 3 summarizes the mean values of each metric for the easiest (bin 1) and hardest (bin 10) examples, averaged across all datasets and seeds. All three metrics display a strong positive correlation with difficulty bin number, indicating that additional training preferentially modifies the predictions of the hardest examples.

### H.3. Analysis

All three metrics exhibit a strong, monotonic increase across the 10 difficulty bins (Pearson correlations between bin number and per-bin metric values: $\Delta m$: $r$=0.914 ($p$=0.0002), $\Delta\ell$: $r$=0.820 ($p$=0.004), predictive shift: $r$=0.920 ($p$=0.0002)). These highly significant correla-

tions demonstrate that difficulty bin systematically predicts the magnitude of model changes during extended training, ruling out random variation as an explanation. This pattern reveals a coherent story:

- **Confidence increases (margin)** for hard examples are more than two orders of magnitude larger than for easy examples (244.6×). Easy examples already achieve near-maximal margins at 256 steps ($m \approx 0.997$), leaving little room for improvement, while hard examples undergo substantial margin increases ($\Delta m = 0.39$) as the model learns to confidently classify them.

- **Loss reductions** from 256 to 2048 steps show the most extreme differential effect (653.5×), indicating that extended training "forces" difficult examples into the correct class by dramatically reducing their cross-entropy loss, even when the top-1 prediction was already correct at the earlier checkpoint.

- **Probability distributions** shift substantially more for

| Metric | Easy (bin 1) | Hard (bin 10) | Ratio (Hard/Easy) |
|---|---|---|---|
| $\Delta m$ (margin) | 0.0016 | 0.3911 | 244.6 |
| $\Delta \ell$ (loss) | 0.0009 | 0.5924 | 653.5 |
| Predictive shift ($L_1$) | 0.0039 | 0.5003 | 129.4 |

*Table 3.* Per-bin memorization indicators averaged across datasets and seeds. Hard examples exhibit dramatically larger changes between the 256- and 2048-step models, consistent with late-stage memorization.

difficult examples (129.4×), suggesting targeted late-stage adjustments to the decision function. While easy examples maintain stable predictions ($L_1$ shift $\approx 0.004$), hard examples experience large redistributions of probability mass across classes ($L_1$ shift $\approx 0.50$).

Together, these metrics provide consistent and quantitative evidence that longer finetuning causes the model to memorize difficult examples: the 2048-step models exhibit large, difficulty-dependent changes to losses, margins, and probability distributions that are absent or minimal in the 256-step models. These memorization patterns have important implications for task vector composition and model merging. Models at different training stages have encoded fundamentally different solutions to the same classification task: early models rely on generalizable features that work across many examples, while late models additionally employ example-specific adjustments that memorize individual difficult cases. When such heterogeneous models are combined via task arithmetic or model merging, the interaction between generalizable and memorized components can produce unexpected emergent behaviors, potentially explaining performance variability in multi-task and few-shot transfer settings.

# I. Quantifying parameter interference throughout training

**Task Vector Construction** For each dataset $d$ and training step $t$, we compute a task vector $\tau_d^t$ as the parameter difference between the fine-tuned model and the pre-trained (zero-shot) model:

$$\tau_d^t = \theta_d^t - \theta_0 \qquad (2)$$

where $\theta_d^t$ represents the parameters of the model fine-tuned on dataset $d$ for $t$ steps, and $\theta_0$ represents the pre-trained model parameters. All floating-point parameters are concatenated into a single vector of dimension $n$.

**Top-$k$% Pruning** For analysis, we apply magnitude-based pruning to the task vectors to retain only the most significant parameter changes. For a given task vector $\tau$ and pruning

threshold $k$%, we define the pruned task vector:

$$\tilde{\tau}_i = \begin{cases} \tau_i & \text{if } |\tau_i| \geq \text{threshold}_k \\ 0 & \text{otherwise} \end{cases} \qquad (3)$$

where $\text{threshold}_k$ is the $k$-th percentile of $|\tau|$. Unless otherwise specified, we use $k = 20\%$ (retaining the top 20% of parameters by absolute magnitude), following the best practice from Yadav et al. (2023).

## I.1. Metric Definitions

**Sign Conflict Percentage.** For each dataset pair $(d_1, d_2)$, we count the proportion of parameters where the task vectors have opposite signs. Let $\mathcal{A} = \{i : \tilde{\tau}_{d_1,i} \neq 0 \vee \tilde{\tau}_{d_2,i} \neq 0\}$ be the set of active parameters, and $\mathcal{C} = \{i \in \mathcal{A} : (\tilde{\tau}_{d_1,i} > 0 \wedge \tilde{\tau}_{d_2,i} < 0) \vee (\tilde{\tau}_{d_1,i} < 0 \wedge \tilde{\tau}_{d_2,i} > 0)\}$ be the set of parameters with conflicting signs. Then:

$$\text{SignConflict}(d_1, d_2) = \frac{|\mathcal{C}|}{|\mathcal{A}|} \times 100 \qquad (4)$$

High sign conflict indicates destructive interference when averaging task vectors.

**Parameter Overlap Percentage** We measure how much the important parameters (top-$k$% by magnitude) overlap between datasets. For each dataset $d$, let $\mathcal{M}_d = \{i : |\tau_{d,i}| \geq \text{threshold}_k\}$ be the set of important parameter indices. The overlap between two datasets is:

$$\text{Overlap}(d_1, d_2) = \frac{|\mathcal{M}_{d_1} \cap \mathcal{M}_{d_2}|}{|\mathcal{M}_{d_1}|} \times 100 \qquad (5)$$

High overlap indicates that different tasks compete for the same parameters, increasing the potential for interference during merging.

**Magnitude Ratio** For parameters that are important in both task vectors (i.e., in the overlapping set $\mathcal{O} = \mathcal{M}_{d_1} \cap \mathcal{M}_{d_2}$), we measure the disagreement in their magnitudes:

$$\text{MagRatio}(d_1, d_2) = \frac{1}{|\mathcal{O}|} \sum_{i \in \mathcal{O}} \frac{\max(|\tau_{d_1,i}|, |\tau_{d_2,i}|)}{\min(|\tau_{d_1,i}|, |\tau_{d_2,i}|)} \qquad (6)$$

A magnitude ratio significantly greater than 1 indicates that even when tasks modify the same parameters, they disagree substantially on the extent of modification, leading to interference when averaged.

**Per-Parameter Variance** Unlike the pairwise metrics above, per-parameter variance captures multi-task disagreement. For each parameter position $i$, we compute the variance of its changes across all $D$ datasets:

$$\text{Var}(i) = \frac{1}{D} \sum_{d=1}^{D} (\tilde{\tau}_{d,i})^2 \qquad (7)$$

where we assume the mean is centered at zero (the pre-trained model). The overall variance metric is:

$$\text{Variance} = \frac{1}{|\mathcal{A}_{\text{all}}|} \sum_{i \in \mathcal{A}_{\text{all}}} \text{Var}(i) \qquad (8)$$

where $\mathcal{A}_{\text{all}} = \bigcup_{d=1}^{D} \{i : \tilde{\tau}_{d,i} \neq 0\}$ is the union of all active parameters across datasets. Higher variance indicates greater disagreement about how each parameter should be modified, leading to information loss during averaging.

### I.2. Results

We analyze ViT-B-32 checkpoints trained on the 8 considered vision datasets (Cars, DTD, EuroSAT, GTSRB, MNIST, RESISC45, SUN397, SVHN) across 6 training step counts (4, 16, 64, 256, 512, 2048) and 3 random seeds.

For pairwise metrics (Sign Conflict, Parameter Overlap, Magnitude Ratio), we:

1. Compute the metric for all $\binom{8}{2} = 28$ dataset pairs at each step and seed

2. Average across the 28 pairs to obtain a single value per step per seed

3. Compute the mean and standard deviation across the 3 seeds

For the per-parameter variance metric, we:

1. Compute variance across all 8 datasets simultaneously at each step and seed

2. Compute the mean and standard deviation across the 3 seeds

The results are presented in Table 4 and Figure 11.

Table 4 presents parameter interference metrics across training steps, revealing systematic increases in all interference measures as training progresses. Sign conflict percentage increases modestly from 7.24% to 8.03%, indicating a slight rise in parameters with opposing signs across tasks. Parameter overlap grows from 26.09% to 29.53%, showing that longer training causes different tasks to increasingly compete for the same important parameters. The magnitude

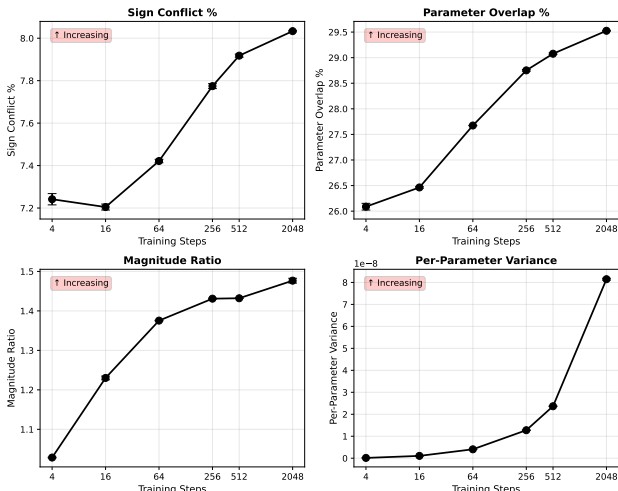

*Figure 11.* Parameter interference metrics across training steps (mean $\pm$ std across 3 seeds). Metrics computed on the union of each dataset's top 20% parameters by magnitude.

ratio increases from 1.03 to 1.48, demonstrating growing disagreement about the extent of parameter modifications even when tasks modify the same parameters in the same direction. Most dramatically, per-parameter variance increases by approximately 60-fold from $1.35 \times 10^{-10}$ to $8.15 \times 10^{-8}$, providing strong evidence that extended training causes datasets to diverge substantially in their parameter modifications. Collectively, these metrics demonstrate that longer training amplifies parameter interference, directly explaining the degradation in merge performance observed with increased training steps. The consistency of these trends across all metrics and the low standard deviations across seeds indicate that this phenomenon is robust and reproducible.

## J. Pseudocode for the ViT early stopping strategy

Here we provide pseudocode for the LR scheduler and early stopping strategy with linear warm-up and adaptive decay used for ViT models.

## K. Model MoErging results

Model MoErging (Yadav et al., 2025) is another popular way to re-use existing model checkpoints. As opposed to model merging, which combines the model parameters directly, MoErging approaches such as Ostapenko et al. (2024); Muqeeth et al. (2024) combine adapters into modular, mixture-of-experts (MoE) type layers (Shazeer et al., 2017) expanding the model's size and capabilities. A routing mechanism determines which input, or part of the input, gets processed by which expert modules. For this upcycling

*Table 4.* Parameter interference metrics across training steps (mean $\pm$ std across 3 seeds). Metrics computed on the union of each dataset's top 20% parameters by magnitude.

| Steps | Sign Conflict (%) | Param Overlap (%) | Mag Ratio | Variance |
|---|---|---|---|---|
| 4 | $7.24 \pm 0.03$ | $26.09 \pm 0.07$ | $1.03 \pm 0.00$ | $(1.35 \pm 0.00) \times 10^{-10}$ |
| 16 | $7.20 \pm 0.01$ | $26.46 \pm 0.02$ | $1.23 \pm 0.00$ | $(1.06 \pm 0.00) \times 10^{-9}$ |
| 64 | $7.42 \pm 0.01$ | $27.67 \pm 0.00$ | $1.38 \pm 0.00$ | $(4.05 \pm 0.01) \times 10^{-9}$ |
| 256 | $7.77 \pm 0.01$ | $28.75 \pm 0.02$ | $1.43 \pm 0.00$ | $(1.27 \pm 0.01) \times 10^{-8}$ |
| 512 | $7.92 \pm 0.01$ | $29.08 \pm 0.02$ | $1.43 \pm 0.00$ | $(2.37 \pm 0.01) \times 10^{-8}$ |
| 2048 | $8.03 \pm 0.00$ | $29.53 \pm 0.01$ | $1.48 \pm 0.01$ | $(8.15 \pm 0.04) \times 10^{-8}$ |

---

**Algorithm 1** Early stopping strategy with linear warm-up and adaptive decay

---

**Input:** Peak LR $\eta_{\max}$, warm-up steps $S_{\text{warm}}{=}50$, validation interval $S_{\text{val}}{=}5$, patience $P{=}3$, decay factor $\gamma{=}0.5$, minimum LR threshold $\eta_{\min} = 1e{-}7$
Initialize learning rate $\eta \leftarrow 0$
Initialize best validation accuracy $A_{\text{best}} \leftarrow 0$
Initialize plateau counter $c \leftarrow 0$
**for** training step $s = 1, 2, \ldots$ **do**
  **if** $s \leq S_{\text{warm}}$ **then**
    *Linear warm-up*
    $\eta \leftarrow \eta_{\max} \cdot \frac{s}{S_{\text{warm}}}$
  **else**
    *Adaptive decay and early stopping*
    **if** $s \bmod S_{\text{val}} = 0$ **then**
      Evaluate validation accuracy $A_{\text{val}}$
      **if** $A_{\text{val}} > A_{\text{best}}$ **then**
        $A_{\text{best}} \leftarrow A_{\text{val}}$
        $c \leftarrow 0$
      **else**
        $c \leftarrow c + 1$
      **end if**
      **if** $c \geq P$ **then**
        $\eta \leftarrow \gamma \cdot \eta$
        $c \leftarrow 0$
      **end if**
      **if** $\eta < \eta_{\min}$ **then**
        **stop training**
      **end if**
    **end if**
  **end if**
  Update model parameters using $\eta$
**end for**

---

strategy further training is often required to let the router and expert adapters learn how to interact with one another. In this section we analyze the impact of overtraining experts on downstream model MoErging.

### K.1. Preliminaries

MoErging methods aggregate multiple fine-tuned experts with the use of modular architectures, such as mixture-of-experts (MoE) layers (Shazeer et al., 2017), to build stronger deep learning systems. The large design space of these methods, paired with their effectiveness has led to the rapid development of many new methods in the recent past, with varying expert, router and application design choices (Yadav et al., 2025; Huang et al., 2024; Ostapenko et al., 2024; Muqeeth et al., 2024). A key feature of MoErging approaches is modularity; multiple experts are considered simultaneously and a routing mechanism decides which input, or part of an input, is processed by which expert.

In this work we consider per-token and per-layer routing, following recent works which suggest this leads to better performance relative to other possible configurations (Ostapenko et al., 2024; Muqeeth et al., 2024). Concretely, let $\mathbf{W} \in \mathbb{R}^{d_{\text{out}} \times d_{\text{in}}}, b \in \mathbb{R}^{d_{\text{out}}}$ denote the weight matrix and bias of a pre-trained linear layer, whose original output is $\mathbf{W}x + b$. We assume the availability of a fine-tuned expert module $E_t(\cdot)$ for each target task $t \in \mathcal{T}$ and we replace the original linear layer with a MoE layer. A router $\pi$ parameterized by matrix $R \in \mathbb{R}^{|\mathcal{T}| \times d_{\text{in}}}$ computes routing logits $Rx$ and applies softmax $\sigma(\cdot)$ to obtain the routing probabilities. The outputs of the experts with top $k$ highest probabilities are then computed and weight-averaged. The resulting MoE layer output is:

$$y = \mathbf{W}x + b + \frac{\sum_{t \in I_k(x)} \pi(x)_t \, E_t(x)}{\sum_{t \in I_k(x)} \pi(x)_t}, \qquad (9)$$

where $I_k(x) = \{t \mid \pi(x)_t \in \text{top k elements of } \pi(x)\}$. We use $k = 2$ for our experiments.

We consider the "multi-task" setting where we assume access to all the datasets the experts were trained on. After updating every linear layer of the pre-trained model with available adapters, we continue training the MoE-fied model on the multi-task mixture of data by freezing the original model parameters and only updating the router and the expert modules. Our MoErging setup is closely related to Ostapenko et al. (2024), which combines LoRA experts

*Table 5.* Early stopping MoErging results

| Expert initialization | Avg. accuracy |
| --- | --- |
| 2048 steps LoRAs | $85.1 \pm 0.1$ |
| Best LoRAs (256 steps) | $87.3 \pm 0.2$ |
| Early stop LoRAs | $87.3 \pm 0.1$ |

into MoE layers and initializes routers using Arrow, but we additionally assume access to the training data and continue multi-task training. Notably, while prior work spans many routing and architectural designs, none study how expert overtraining affects downstream MoErging performance.

**Model MoErging with LoRA adapters**  Using LoRA adapters for model MoErging is straight-forward, with each adapter being used to define one expert module in the MoE layer. Let $\mathbf{A}_t$ and $\mathbf{B}_t$ denote the LoRA low-rank matrices obtained from fine-tuning on task $t$, then we can define the expert modules in Equation (9) as $E_t(x) = \frac{\alpha}{r}\mathbf{B}_t\mathbf{A}_t x$ for each task of interest $t \in \mathcal{T}$.

### K.2. Expert overtraining hurts Model MoErging

We now analyze how the performance of MoE-fied models, initialized with LoRA experts, is affected by the training time of these experts. We use LoRA adapters trained on each of the 8 classification tasks (see Section 3 for details) for different numbers of training steps to initialize our MoE experts, one LoRA for each task. The routing mechanism is initialized using Arrow (Ostapenko et al., 2024), where the weight vector associated with each expert is the first right-singular vector of the $BA$ matrix multiplication. These vectors are assumed to determine the direction of most variance induced by expert $E_t$ for $t \in \mathcal{T}$ in the space of hidden states induced by data from task $t$.

We create one MoE-fied model for each number of steps $s$, i.e. for each different model we initialize the MoE layers with the expert LoRAs for each task, all trained for $s$ steps. Once the MoE-fied model has been initialized using the fine-tuned LoRAs, we further train the routing mechanism and the LoRA experts in a multi-task fashion for 4000 steps with a peak learning rate of 1e-5, with the base parameters frozen. We report the final, multi-task, accuracies over the 8 classification tasks in Figure 12.

We observe that the MoE-fied models initialized with over-trained LoRA experts reach about 2% lower final multi-task accuracy than the models initialized with experts trained for less. Even expert LoRAs trained for as little as 4 steps on their respective tasks reach a higher final multi-task accuracy than those overtrained. We conclude that overtraining experts can hurt downstream MoErging.

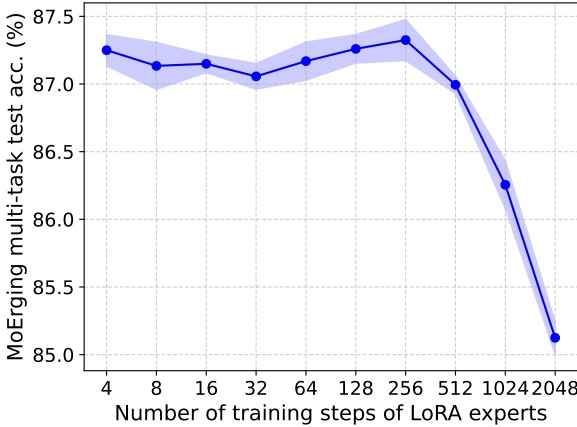

*Figure 12.* Final multi-task accuracy of the MoE-fied models as a function of the number of training steps used to obtain the LoRA experts used for initializing the MoE-fied model. Mean and standard deviation across 3 different initializations of the experts are shown.

### K.3. Early stopping

We also use the early-stopped LoRAs, selected via the early stopping criterion described in the Section 5, to initialize MoE layers and continue training in a multi-task fashion, as in the previous section. As shown in Table 5, the MoErged models initialized with the early stop LoRAs achieve the same accuracy as the best LoRAs across all training steps.

