# OpenReview forum: "From Memorization to Parameter Interference: How Overtraining Experts Harms Model Merging"
_ICML.cc/2026/Conference — ICML 2026 regular_

### Official Review · Reviewer_3SmD · 2026-02-16

**Soundness:** 3
**Presentation:** 3
**Significance:** 1
**Originality:** 2
**Overall Recommendation:** 3
**Confidence:** 5

**Summary:**

This paper studies how the duration of task-specific finetuning affects downstream model merging and MoErging. Contrary to the common assumption that better experts yield better merges, the authors show that longer finetuning often hurts merged performance across vision and NLP tasks, both in full finetuning and PEFT, and across multiple merge methods. They trace the degradation to late-stage memorization of a small set of hard examples that induces idiosyncratic parameter updates and negative interference, and demonstrate that task-dependent aggressive early stopping reliably improves merging quality.

**Compliance With Llm Reviewing Policy:**

Affirmed.

**Final Justification:**

The paper is very insightful and it deeply analyzes the phenomenon "overtraining hurts model merging". The main discovery and practical takeaway of the paper are that **undertrained checkpoints are more mergeable, and checkpoints early in their optimization trajectory should be released for practitioners to perform better model merging**. However, this particular phenomenon and take-home message were already discovered and proposed in previous work, although under a different and more theoretical perspective. Therefore, I view the main novel contribution of this paper to be the more empirical analysis associating this phenomenon with the memorization of hard samples during finetuning.

Overall, I find the discovery of the "undertraining benefits model merging" to be not novel, but I do appreciate the comprehensive empirical analysis, which alone I believe is still not sufficient for ICML publication.

**Key Questions For Authors:**

1. For early stopping strategies, what validation signals (e.g., plateau patience, LR thresholds) worked best across vision vs. NLP, and how sensitive are merging gains to hyperparameter choices like patience=3?

2. Do you observe any cases where training longer helps merging (e.g., very small datasets or highly aligned tasks)? If so, can your interference metrics or diagnostics predict these exceptions?

**Limitations:**

The paper only tested four merging methods and basic LoRA. To make the results more convincing and universal, additional algorithms should be evaluated. I suggest adding TSV and Isotropic merging, which are based on low-rank properties of task vectors (unexplored in this paper yet). Although early-stopping can yield more mergeable checkpoints, the downside of diminished task performance is an issue. I encourage the authors to further explore the tradeoff between mergeability and task performance. Once the tradeoff has been found, a more practical strategy might emerge.

**Strengths And Weaknesses:**

## Strengths
1. The paper examines a timely and clear problem: how finetuning affects model merging and MoErging.
2. Provides a clear, empirically grounded explanation of why excessive finetuning hurts mergeability and MoErgeability.
3. Further explains the phenomenon through the lens of data difficulty, showing that memorizing difficult samples incurs parameter interference when merging.
4. Proposes a simple, practical early-stopping strategies (task-dependent) that consistently improve merged accuracy with less compute.
5. Broad, systematic sweeps over training duration for both FFT and LoRA experts; evaluation across four popular merge methods, two modalities, and multiple model sizes.
6. Actionable guidance for the community: publish intermediate checkpoints and prefer early-stopped experts for model merging.



## Weaknesses
1. The main contribution is somehow incremental: The insight that overtraining hurts model merging, although explained through data difficulty, is not novel, and has already been identified in several works [1, 2, 3, 4], one of which you cited in the paper, but the other two are missing.
2. Because task-dependent early stopping improves mergeability at the cost of task-specific performance, it is not of practical interest for users who are primarily concerned with task performance rather than future mergeability. Hence, the proposed early-stopping method is not practically appealing in general, unless model merging is the end goal of the user during finetuning.
3. Limited merging algorithms: The authors examine four common merging algorithms, but they are not representative enough. Some representative SOTA methods are missing, for example, TSV and Isotropic merging, which are based on completely different assumptions.


[1] On Task Vectors and Gradients: https://arxiv.org/pdf/2508.16082

[2] Twin-Merging: Dynamic Integration of Modular Expertise in Model Merging: https://arxiv.org/pdf/2406.15479

[3] Less is More: Undertraining Experts Improves Model Upcycling: https://arxiv.org/pdf/2506.14126

[4] ATM: Improving Model Merging by Alternating Tuning and Merging: https://arxiv.org/pdf/2411.03055

---

> ### Author Rebuttal · Authors · 2026-03-31
>
> We thank the reviewer for the positive remarks and address the concerns below.
>
> ### **Novelty relative to past works**
> We acknowledge that prior works have observed or mentioned this phenomenon, however they do not validate or analyze it in depth. (Pari et al., 2024) identify representational incompatibility (via CKA) in a small, two-expert setting and propose bypassing merging altogether in favor of MoE routing. (Zhou et al., 2025 [1,4]) observe that early task vectors improve task arithmetic, but do not analyze why; they propose alternating training and merging steps, which is unrealistic in most merging scenarios and closer to distributed training. Our contributions go substantially beyond these observations:
>
> 1. **Validation across settings:** We demonstrate the phenomenon across 2 fine-tuning strategies, 2 upcycling methods, and 2 domains/architectures, as well as many merging methods.
> 2. **Mechanistic explanation:** We explain the phenomenon via the memorization of hard examples, linking our results to the rich literature on data difficulty.
> 3. **Practical mitigation:** We provide a simple, effective solution through early stopping.
>
> The breadth of validation, the mechanistic grounding, and the actionable mitigation distinguish our work from prior observations.
>
> ### **On the significance of our work**
> We respectfully disagree with the low significance assessment. ICML 2026 explicitly solicits “original and rigorous research of significant interest to the machine learning community”, including "evaluation (methodology, meta studies, replicability and validity, human-in-the-loop, etc.)." Our paper is exactly such a contribution: an empirical study of a previously underappreciated failure mode in model merging, robust across diverse settings and practically actionable.
>
> Model merging and MoErging have exploded in popularity, yet nearly all prior work focuses on designing new merging algorithms rather than analyzing how standard fine-tuning itself affects mergeability. Our findings overturn a widely assumed but untested default and fill an important gap in the literature by providing a new axis for improving upcycling that is **orthogonal to method design**. A key outcome is that **choosing the right checkpoint can matter more than choosing the right merging method: merging undertrained experts can outperform advanced algorithms applied to overtrained ones** (lines ~169–171 right column and Figure 2 left). This implies that much "merging difficulty" originates upstream in the experts, not downstream in the method.
>
> ### **On the practical interest of our work**
> We thank the reviewer for raising this point. We agree that if a practitioner's sole objective is maximizing standalone expert performance with no intention of ever merging or composing models, then mergeability is not a very relevant criterion. However, in the increasingly common setting where experts are intended for downstream composition and upcycling in multi-task systems, our finding that "natural" checkpoint selection (i.e., later checkpoints) can be harmful for downstream merging is an important and actionable insight.
>
> Moreover, even for practitioners who are not primarily focused on merging, our recommendations remain practical and low-cost. In particular, publishing intermediate checkpoints requires negligible additional effort yet substantially increases the usefulness of released models for future reuse, analysis, and composition by others. This aligns with an emerging trend in the community toward releasing checkpoints along the training trajectories rather than only the final ones, and represents a simple, actionable guideline with high potential positive impact for the broader ecosystem.
> ### **Additional merging methods**
> We agree that including stronger and more diverse baselines is important, therefore we have conducted experiments with additional SVD-based merging methods such as TSV and Iso-C. Please see our response to reviewer nx4x for these results, the same phenomenon can be observed with these more advanced methods.
>
> ### **Question 1**
> Across our experiments, the gains from early stopping are robust to exact hyperparameter choices, provided stopping is aggressive enough to avoid the late-stage regime where mergeability deteriorates. Reasonable hyper-parameter values perform similarly, with values resulting in longer training performing worse; we can add a clearer robustness summary in the revision.
>
> ### **Question 2**
> We did not observe settings where longer expert training improved merged performance once experts had passed the best mergeable checkpoint, even when longer training continued improving standalone performance. Our analysis confirms later gains involve expert-specific updates that amplify interference after merging. While we cannot rule out exceptions in other settings, none arose in our experiments.

---

> > ### Author Rebuttal · Reviewer_3SmD · 2026-03-31
> >
> > Many thanks to the authors for preparing the rebuttal. I really appreciate the work itself.
> >
> > The rebuttal clarified my questions, but did not address my fundamental concern.
> >
> > ## Biggest Concern: Novelty
> > My biggest concern is **novelty**. I agree with the authors that they analyzed the phenomenon of "overtraining hurts merging" very much in depth, providing a different perspective on the phenomenon. However, the phenomenon would be new if this paper were submitted two years ago. Right now, it is really not new, and this paper just provides yet another analysis. I admit the analysis is deep and carried out well.
> >
> > The paper [3] already provided empirical evidence of this phenomenon, while [1] DID explain why early task vectors merge better from the perspective of gradient fidelity, even providing mathematical proofs. Since in this paper the authors also mostly used task-vector-based merging methods, the analysis from [4] applies here as well.
> >
> > [1] On Task Vectors and Gradients (https://arxiv.org/pdf/2508.16082)
> >
> > [3] Less is More: Undertraining Experts Improves Model Upcycling (https://arxiv.org/pdf/2506.14126)
> >
> > ## On Significance
> > I respectfully clarify that I do not want to degrade the significance of this work; it is interesting by nature. I only stated that the main observation is already observed and analyzed (differently) in several works, and having to do early stopping and upload intermediate checkpoints is not of practical interest to most practitioners who do not care about model merging. That said, I wholeheartedly support the idea of releasing intermediate checkpoints, which would be really great. However, the overhead is definitely not negligible; it requires storing and uploading multiple checkpoints.

---

> > > ### Author Response · Authors · 2026-04-02
> > >
> > > ### **On novelty**
> > > We would like to thank the reviewer for the fast response and for fully engaging with our rebuttal. First off we would like to point out that [1] and [4] are two different versions of the same work, or at least highly related works from the same authors, with very similar ideas. [1] is a workshop paper and focuses on theory while [4] is an arXiv preprint presenting the ATM framework for alternating training and merging, both discuss the relation of task vectors to the multi-task gradients. We have already addressed [4] in our paper and in the initial rebuttal, their analysis is minimal and insufficient when compared to ours and their proposed ATM method is not realistic in actual merging settings. We’ll therefore only address [1] here since it formalizes the ideas and intuitions from [4] with their theoretical results.
> > >
> > > We agree that the theoretical results presented in [1] are valuable, and their analysis from the perspective of gradient fidelity is interesting. However, we view their work as complementary to ours. Deep learning as a field has been characterized by strong empirical results on one side and theoretical understanding on the other. It is not uncommon for theoretical papers not to receive significant attention because their results are limited to specific, often idealized settings and the complexity of modern deep learning pipelines makes it unclear to what extent the theoretical results extend to more realistic scenarios. **We highlight some limitations of the theoretical results from [1]:**
> > > - The authors themselves acknowledge that their analysis is limited by their full-batch Gradient Descent setting, which can be quite different from SGD and modern optimizers used in practice such as AdamW.
> > > - Furthermore, some of their theoretical results are only established for feedforward networks which, again, are not representative of model architectures used in practice.
> > > - Lastly, their analysis concerns entire epochs of training, which we argue is not always a good proxy for training progress and can be somewhat arbitrary. Depending on the size and the complexity of the dataset, one epoch of training might not be enough or might be too much. In our own experiments, for instance, a dataset such as MNIST which is simple and has a lot of examples requires significantly less training steps (way less than an entire epoch) for good generalization. Other datasets, that have less examples but represent more complex tasks, might require multiple epochs of training before good generalization.
> > >
> > > We do not highlight these limitations to critique [1], we know establishing such theoretical results can be hard even in idealized settings and sometimes intractable in realistic settings. We therefore salute their efforts and believe that their analysis at the very least provides useful intuition about this phenomenon. However, **there often is a significant gap between theoretical results and what can reliably be observed empirically.** Our work fills this gap, we consider modern model architectures and training pipelines and our results show concretely the merging performance decrease that can occur due to prolonged training. This is of significant interest to practitioners who might want to know how much their fine-tuning procedure can affect their merging results. This represents a concrete and direct addition to [1] and past works. In this sense, we view our work as complementary to [1], not conflicting in terms of novelty.
> > >
> > > We hope this clarifies the novelty concerns regarding [1] and [4]. If any novelty or significance concerns remain, are they primarily about the relation to [3]? If so, we respectfully suggest a careful comparison of the two works.
> > >
> > > ### **On significance**
> > > We thank the reviewer for acknowledging the utility of releasing intermediate checkpoints. We agree that the overhead is not necessarily “negligible” but it is also not a significant issue. Those who have the computational ressources to train a model, typically have the storage ressources to preserve one or two additional checkpoints along the training trajectory. In fact, checkpointing during training is already standard practice, a small number of those checkpoints then only need to be released. Furthermore we don’t suggest people change their training procedures to implement early stopping for merging purposes, earlier checkpoints from the standard training procedure should be sufficient to see these merging improvements.

---

### Official Review · Reviewer_nx4x · 2026-03-12

**Soundness:** 3
**Presentation:** 3
**Significance:** 2
**Originality:** 2
**Overall Recommendation:** 5
**Confidence:** 5

**Summary:**

This paper perform comprehensive experiments across vision and language tasks, and both full fine-tuning and LoRA settings to show that training expert models longer before merging them actually produces worse merged models, even though the individual experts get stronger. The authors explored its connection with hard examples and propose task-dependent early stopping as a simple fix.

**Compliance With Llm Reviewing Policy:**

Affirmed.

**Key Questions For Authors:**

In Figure 1 left, I guess the base model performance is around 55% accuracy? What's the reason TIES and DARE fall below that level at very few training steps?
After TIES and DARE, there have been other merging methods aiming to reduce inference, as well as short multitask training after averaging, a trick used in Command A. I wonder, does any of them shift the optimal time to stop?

**Limitations:**

Yes.

**Strengths And Weaknesses:**

Strength:
- Well supported main argument: The authors provide through experiments with FFT / LoRA, multiple merging methods and a dense grid of steps.  The curves clearly show how number of steps affect interference.

Weakness:
- Systematic study of known phenomena: As the authors also have explained in related works, Pari, J et al ready demonstrated the main finding of this paper at small scale.
- Uncertainty about hard examples: Delving into hard example didn't lead to a clear understanding of the mechanism of increased interference in prolonged finetuning. Hard examples drives a longer period of training, and yet at the same time improves merging performance. (Nitpicking. The experiments on example difficulty is still a nice addition.)

---

> ### Author Rebuttal · Authors · 2026-03-31
>
> We thank the reviewer for highlighting the strength and breath of our empirical results. We address the reviewer’s concerns below:
>
> ### **Novelty relative to past works**
> We acknowledge that prior works have observed or mentioned this phenomenon, however they do not validate or analyze it in depth. (Pari et al., 2024) identify representational incompatibility (via CKA) in a small, two-expert setting and propose bypassing merging altogether in favor of MoE routing. (Zhou et al., 2025) observe that early task vectors improve task arithmetic, but do not analyze why; they propose alternating training and merging steps, which is unrealistic in most merging scenarios and closer to distributed training. Our contributions go substantially beyond these observations:
>
> 1. **Validation across settings:** We demonstrate the phenomenon across 2 fine-tuning strategies, 2 upcycling methods (merging and MoErging), 2 domains (vision and language), multiple architectures and model sizes, as well as many SOTA merging methods.
> 2. **Mechanistic explanation:** We explain the phenomenon via the memorization of hard examples, linking our results to the rich literature on data difficulty.
> 3. **Practical mitigation:** We provide a simple, effective solution through early stopping.
>
> The breadth of validation, the mechanistic grounding, and the actionable mitigation clearly distinguish our work from prior observations.
>
> ### **Uncertainty about hard examples**
> We believe the reviewer’s comment regarding the role of hard examples highlights a real tension, rather than a contradiction. Hard examples can benefit both individual model performance and merging, as our results in Figure 3 suggest, **but only up to a point**. This is analogous to standard overtraining: difficult examples can improve generalization early on, yet prolonged training on them can eventually hurt performance. In our setting, extended training appears harmful when experts begin to fit hard examples in increasingly idiosyncratic ways. Our recommendation is therefore not to avoid hard examples, but to avoid overtraining expert models.
>
> ### **Question: Figure 1**
> The base model performance is in fact around 49%, not 55%. Average expert performance rises quickly from 49% at initialization to ~55% after two fine-tuning steps and to ~60% after four steps. So TIES and DARE at early checkpoints are not falling below the base model; rather, they are underperforming relative to the average of the individually fine-tuned experts.
>
> ### **Question: other merging methods**
> We agree this is an important question. In response, we ran additional experiments with more recent SVD-based methods, including TSV [1] and ISO-C [2], whose mechanisms differ substantially from prunning-based methods such as TIES or DARE. See the table below for the results. The same phenomenon can be observed: prolonged expert fine-tuning degrades merging performance for the SVD-based methods as well, and the optimal merge point still occurs earlier than the optimal standalone-expert checkpoint. This strengthens our claims and establishes their validity even beyond pruning-based merging techniques. We will be incorporating these results into the camera ready version to strengthen the paper.
>
> | Method | 63 | 127 | 255 | 511 | 1023 | 2047 |
> | ------- | ---------: | ---------: | ---------: | ---------: | ---------: | ---------: |
> | Iso-C | 70.3 ± 0.3 | 74.2 ± 0.0 | 76.3 ± 0.3 | 76.2 ± 0.2 | 74.6 ± 0.6 | 74.1 ± 0.4 |
> | Iso-CTS | 69.7 ± 0.5 | 72.4 ± 1.0 | 75.6 ± 0.2 | 75.8 ± 0.5 | 73.9 ± 0.2 | 73.3 ± 0.7 |
> | TSV-M | 69.7 ± 0.3 | 76.1 ± 0.1 | 78.8 ± 0.2 | 78.6 ± 0.2 | 76.7 ± 0.8 | 76.4 ± 0.6 |
>
> ### **Question: short multi-task training**
> We agree that post-merge multitask training is an interesting direction. However, it introduces an additional optimization stage and adds confounding factors relative to the central question of our paper: how the pre-merge training of experts affects mergeability. For this reason, we consider it outside the scope of our work. That said, our MoErging results in Appendix K already include a multitask-training phase after combining experts, and the same conclusion holds there: overtraining the constituent experts still hurts final upcycling performance. We would expect similar results to hold for multi-task training after model merging.
>
> ### References
> [1] Antonio Andrea Gargiulo, Donato Crisostomi, Maria Sofia Bucarelli, Simone Scardapane, Fabrizio Silvestri, Emanuele Rodolà. *Task Singular Vectors: Reducing Task Interference in Model Merging*. CVPR 2025
> [2] Daniel Marczak, Simone Magistri, Sebastian Cygert, Bartłomiej Twardowski, Andrew D. Bagdanov, Joost van de Weijer. *No Task Left Behind: Isotropic Model Merging with Common and Task-Specific Subspaces*. ICML 2025

---

> > ### Author Rebuttal · Reviewer_nx4x · 2026-04-02
> >
> > Thanks for clarification and added experiments. My concerns are resolved.

---

### Official Review · Reviewer_4Gc8 · 2026-03-16

**Soundness:** 3
**Presentation:** 3
**Significance:** 2
**Originality:** 2
**Overall Recommendation:** 3
**Confidence:** 3

**Summary:**

The paper studies how the way expert models are trained affects the quality of model merging. Surprisingly, it finds that training experts longer, although it improves their individual performance, can actually hurt the performance of the merged model. The authors show that during the later stages of training, experts tend to memorize harder examples in a way that creates conflicting parameter updates, which leads to interference when models are merged. Based on this observation, they suggest that stopping expert training earlier can improve the performance of merged models.

**Compliance With Llm Reviewing Policy:**

Affirmed.

**Key Questions For Authors:**

You recommend shorter training, while also arguing that the later stages of training focus on learning harder examples. At the same time, you suggest that merging benefits from knowledge about hard examples. Isn’t this somewhat contradictory? If late training learns hard examples and those examples contain useful knowledge, why stop early?

Why do you think in current research people overtrain the experts? Why do they choose the parameters you indicated?

**Limitations:**

The paper focuses on technical aspects of training and merging models. The discussion of limitations appears adequate.

**Strengths And Weaknesses:**

**Strengths:**

The paper identifies an empirical observation: training experts longer improves their individual performance but can harm the performance of merged models. This finding challenges an implicit assumption in many model merging workflows and could influence best practices in training pipelines. it is fairly unintuitive and practically important finding.

They find out and attribute the problem to training hard examples which lead to parameter interference.

They propose a mitigation to the problem of overtraining, early stopping.

The experiments appear to span multiple settings, including both full fine-tuning and LoRA adaptation, domains: vision and language models, and different model sizes.

The paper reads well.


**Weaknesses:**


The paper attributes the degradation in merging performance to memorization of difficult examples during late training stages, leading to parameter interference during merging. However, the evidence supporting this causal explanation is limited. Additional analysis (e.g., memorization metrics, gradient alignment, or parameter-space diagnostics) would strengthen the claim.

While the empirical phenomenon is interesting, the work mainly documents an effect rather than proposing a new method or providing a strong theoretical explanation. The suggested mitigation, early stopping, is simple and somewhat expected once the phenomenon is observed.

The main contribution is the empirical observation that longer fine-tuning harms merging performance. While interesting, the conceptual novelty is somewhat limited. The proposed solution (early stopping) is simple and somewhat expected once the phenomenon is observed. The work therefore reads more as a study than a methodological advance.

---

> ### Author Rebuttal · Authors · 2026-03-31
>
> We thank the reviewer for the positive comments about our work, we address the concerns below.
> ### Memorization and parameter interference
> We agree that the paper's main contribution is the empirical finding. That said, the paper already includes substantial supporting evidence linking prolonged training to memorization and parameter interference. Appendix H reports complementary memorization proxies across training duration, including margin, loss, and predictive distribution shifts between checkpoints, which consistently suggest increased memorization with longer training. Appendix I analyzes parameter-space interference using sign conflicts, parameter overlap, magnitude ratios, and per-parameter variance, all of which increase at later training stages.
>
> Thus, both predictive and parameter-space analyses support our interpretation, though we acknowledge that neither memorization nor parameter interference admits a single definitive metric. We are happy to run any additional diagnostics the reviewer considers suitable here or to adjust our wording.
> ### Regarding early stopping
> We agree that early stopping is simple. However, the key point is precisely “once the phenomenon is observed”: before our study, the natural expectation was that training stronger experts would improve, not hurt, merged performance. That early stopping seems natural in hindsight does not make the finding obvious.
> Our main contribution is systematically identifying that overtraining experts can degrade upcycling performance even while improving individual expert performance. We view this as a strength: it turns a surprising failure mode into a simple, actionable guideline. Moreover, this mitigation is not entirely trivial in practice, since experts may have substantially different optimal training durations leading to seemingly “heterogeneous” merges (as reflected by the high variance in Table 2).
> ### On the significance of our work
> The reviewer notes that "the work reads more as a study than a methodological advance," but we believe this should not count against our paper. ICML 2026 explicitly solicits “original and rigorous research of significant interest to the machine learning community”, including "evaluation (methodology, meta studies, replicability and validity, human-in-the-loop, etc.)." Our paper is exactly such a contribution: an empirical study of a previously underappreciated failure mode in model merging, robust across diverse settings and practically actionable.
>
> Model merging and MoErging have exploded in popularity, yet nearly all prior work focuses on designing new merging algorithms rather than analyzing how standard fine-tuning itself affects mergeability. Our findings overturn a widely assumed but untested default and fill an important gap in the literature by providing a new axis for improving upcycling that is **orthogonal to method design**. A key outcome is that **choosing the right checkpoint can matter more than choosing the right merging method: merging undertrained experts can outperform advanced algorithms applied to overtrained ones** (lines ~169–171 right column and Figure 2 left). This implies that much "merging difficulty" originates upstream in the experts, not downstream in the method.
> ### Question 1 - Hard examples
> We do not believe these claims are contradictory; rather, they reflect a real tension. Hard examples can benefit both individual model performance and merging, **but only up to a point**. This is analogous to standard overtraining: difficult examples can improve generalization early on, yet prolonged training can eventually hurt performance. In our setting, extended training appears harmful when experts begin to fit hard examples in increasingly idiosyncratic ways. Our recommendation is therefore not to avoid hard examples, but to avoid overtraining expert models to reduce cross-expert interference.
> ### Question 2 - Overtrained experts
> We believe this reflects how model merging research has developed. The literature has focused on improving merging algorithms while treating fine-tuning as a fixed step, with training choices like the number of steps adopted by convention rather than examined for their effect on mergeability.
> Importantly, the experts we study are not necessarily overtrained in the classical sense of hurting their own test performance. Rather, they are overtrained for merging: longer fine-tuning can continue to improve individual expert performance while degrading merged model performance (lines 71–74 left column). This makes prior choices understandable: if experts are optimized only for individual task performance, later checkpoints are natural. But it also makes our result especially important, since it shows that a seemingly reasonable checkpoint selection can significantly hurt merging.
> ### New experiments
> We also ran new experiments with modern SVD-based merging methods such as TSV and Iso-C and observed the same phenomenon; see our response to reviewer nx4x.

---

> > ### Author Rebuttal · Reviewer_4Gc8 · 2026-04-04
> >
> > Thank you for your replies. The claims that "the experts we study are not necessarily overtrained in the classical sense of hurting their own test performance. Rather, they are overtrained for merging." and the mechanistic explanation through the hard examples are very valuable. But some better quantification would be useful. This work is promising but I will maintain my score.

---

> > > ### Author Response · Authors · 2026-04-07
> > >
> > > We thank the reviewer for acknowledging the value of our work. To address the remaining concerns about quantifying memorization, we ran additional experiments, described below.
> > >
> > > ### Memorization metrics
> > >
> > > To support our claim that hard-example memorization grows with training duration, we ran experiments inspired by the seminal work on memorization [1, 2]. For each vision dataset, we trained 40 models with different random seeds. In each run, we held out 90 training examples: 30 from the 100 easiest, 30 from the 100 hardest, and 30 from the 100 closest to the median difficulty (as ranked by EL2N scores). Checkpoints were saved every 256 steps, up to 2048 steps total. Under this design, each example in the three bins (easiest, middle, hardest) has an inclusion rate of ~70% across runs. This lets us compute the (Feldman) memorization score [2] for each example $x$ at each checkpoint $t$:
> > >
> > > mem($x$, $t$) = mean(confidence on correct class at step $t$ | $x$ was in training) − mean(confidence on correct class at step $t$ | $x$ was excluded from training)
> > >
> > > **This provides a direct, causal way to quantify memorization across training steps: how much more confident is the model on example $x$ when it was trained on it versus when it wasn't?** A larger gap means including $x$ had more influence on the model's behavior, indicating memorization. We aggregate these scores across datasets and report mean and std below, as a function of training duration and bin.
> > >
> > > | Difficulty bin |          256 |          512 |          768 |         1024 |         1280 |         1536 |         1792 |         2048 |
> > > | -------------- | -----------: | -----------: | -----------: | -----------: | -----------: | -----------: | -----------: | -----------: |
> > > | Easy           | 0.00 ± 0.01 | 0.00 ± 0.01 | 0.00 ± 0.01 | 0.00 ± 0.00 | 0.00 ± 0.00 | 0.00 ± 0.00 | 0.00 ± 0.00 | 0.00 ± 0.00 |
> > > | Middle         | 0.05 ± 0.12 | 0.06 ± 0.13 | 0.06 ± 0.14 | 0.05 ± 0.14 | 0.05 ± 0.14 | 0.05 ± 0.14 | 0.05 ± 0.14 | 0.05 ± 0.14 |
> > > | Hard           | 0.21 ± 0.28 | 0.32 ± 0.32 | 0.38 ± 0.35 | 0.41 ± 0.36 | 0.43 ± 0.39 | 0.44 ± 0.40 | 0.45 ± 0.41 | 0.45 ± 0.41 |
> > >
> > > **Three clear patterns emerge:** (1) Easy examples show zero memorization throughout training (all ≤ 0.01). (2) Middle examples show small, stable memorization (~0.05) that does not grow with training steps. (3) Hard examples show substantial memorization that accumulates throughout training, more than doubling from 0.21 at step 256 to 0.45 at step 2048.
> > >
> > > **These results reinforce our analysis.** Memorization is concentrated on hard examples and grows with training duration, while easy and middle examples remain essentially unaffected. **This shows that later training steps update model parameters primarily to accommodate hard examples:** when a hard example is held out, the model's confidence on it drops sharply, linking its presence in training directly to the model's behavior on it, while holding out easy or middle examples barely changes the model's response.
> > >
> > > We also measure loss-based membership inference AUC [3] at each checkpoint. This provides a complementary privacy-leakage signal related to memorization. The MIA results show the same qualitative pattern: easy examples remain near chance (AUC ≈ 0.5) throughout training, whereas hard examples become increasingly distinguishable, with AUC rising from 0.67 at step 256 to 0.81 at step 2048. In addition, per-example Feldman memorization scores are strongly correlated with a per-example membership signal derived from the loss gap between models trained with and without the example (r>0.81, pooled after within-dataset normalization). Thus, the two analyses are highly consistent and support the same conclusion: hard examples are the ones most strongly memorized as training proceeds.
> > >
> > > **To our knowledge, this is the most direct and principled methodology available for quantifying the relationship between memorization and training duration at the example level.**
> > >
> > > Combined with the parameter interference analysis in Appendix I and the merging results in the main paper, these findings directly support our central claim: longer training, driven primarily by the memorization of hard examples, produces parameter updates that negatively interfere during merging, thus degrading final performance.
> > >
> > > We hope these additional results address the reviewer’s concern, and we would be grateful if they were taken into account in the final assessment.
> > >
> > > ### References
> > > [1] V. Feldman & C. Zhang. “*What Neural Networks Memorize and Why: Discovering the Long Tail via Influence Estimation*”. NeurIPS 2020
> > >
> > > [2] V. Feldman. “*Does Learning Require Memorization? A Short Tale about a Long Tail*”. URL: http://arxiv.org/abs/1906.05271
> > >
> > > [3] S. Yeom, I. Giacomelli, M. Fredrikson & S. Jha. "*Privacy Risk in Machine Learning: Analyzing the Connection to Overfitting*". IEEE CSF 2018

---

### Decision · Program_Chairs · 2026-04-30

**Decision:**

Accept (regular)

**Comment:**

This paper is about model merging. The big question in this area is which models can be merged and which ones cannot. The authors provide solid empirical evidence that specialization via fine-tuning can hurt the ability to merge. They further dig in to observe that the real aspect of specialization that hurts merging is memorizing certain challenging examples. This produces a simple fix that improves mergeability by doing early termination.

These results are interesting and practically valuable, especially considering how important model merging has become. The evidence provided and experiments are pretty comprehensive; the authors do both full fine-tuning and LoRA, several merging methods, etc.

Reviewers were mixed-to-positive, but all reviewers agreed on the promise of the work. Some of the concerns raised: i) the paper is mostly analytical rather than providing algorithmic contributions. This is not a problem, as the analysis here is well-done and valuable. ii) the paper’s key finding has been (in other more limited forms) found in some earlier work. I also do not view this as a major problem, as these earlier works are clearly differentiated (the authors’ rebuttal makes the case in a very reasonable way) and more limited. More generally, I believe a high quality empirical analysis like the one here is valuable and worth including.

Overall this is a strong and practically valuable contribution.